# Anatomy of the Joints in the Hamadryas Baboon (*Papio hamadryas*)—Part 1: Thoracic Limb

**DOI:** 10.3390/ani15192894

**Published:** 2025-10-03

**Authors:** Jolien Horemans, Arthur Fets, Hedwig Donga, Jaco Bakker, Christophe Casteleyn

**Affiliations:** 1Anicura Dierenkliniek De Ark, Probastraat 1, 2235 Westmeerbeek, Belgium; jolien.horemans@anicura.be; 2Department of Morphology, Imaging, Orthopedics, Physiotherapy and Nutrition, Faculty of Veterinary Medicine, Ghent University, Salisburylaan 133, 9820 Merelbeke-Melle, Belgium; arthur.fets@ugent.be (A.F.); hedwigdonga@hotmail.com (H.D.); 3Animal Science Department, Biomedical Primate Research Centre, Lange Kleiweg 161, 2288 GJ Rijswijk, The Netherlands; bakker@bprc.nl; 4Department of Veterinary Sciences, Faculty of Pharmaceutical, Biomedical and Veterinary Sciences, University of Antwerp, Universiteitsplein 1, 2610 Wilrijk, Belgium

**Keywords:** anatomy, topography, hamadryas baboon, arthrology, thoracic limb, joints

## Abstract

The mantled baboon (*Papio hamadryas*) can be found in zoos and research facilities worldwide. When they are injured and need medical care, knowledge of its anatomy is a prerequisite. Unfortunately, finding the anatomical data of interest in the literature is hampered by its scattered presence in often outdated publications. The skeleton of the mantled baboon was described recently. Now, the joints are portrayed. This first work in a series of three discusses all the ligaments and associated structures of the joints of the front limb. The muscle tendons and the bone structures that are related to each joint are also discussed to offer a contextual approach. Since detailed color photographs of dissections support the main text, this manuscript can serve as a dissection guide.

## 1. Introduction

Hamadryas or mantled baboons (*Papio hamadryas*) can be encountered as free-ranging animals in the horn of Africa and the Arabian Peninsula. There, these non-human primates are also kept as pets by nomads. They can, however, be admired worldwide in zoos and are housed in research facilities as human models [1]. It is obvious that captive settings restrict their natural social and foraging behavior and physical needs to prevent stress and abnormal behaviors [2]. Providing these animals with the appropriate diet, social structure, habitat, and medical care is, thus, essential. Regarding the latter, baboons living in captivity might require medical attention when injured. The larger majority of injuries occur in males and are inflicted by means of the canines [3]. The bodily distribution of injuries in yellow baboons (*Papio cynocephalus*) is characterized by a preponderance of injuries at the head, neck, shoulders, chest, and arms [3]. The location of injuries in male hamadryas baboons differs from that of yellow baboons in that most injuries are inflicted on the hands and forearms [4]. At these locations, fractures often occur in the phalanges of the fingers, but fractures of the long bones and the clavicles are also common [5].

Without a doubt, proficiency in species-specific anatomy is fundamental when treating wounded animals. The interpretation of radiographs and the execution of operations illustrate this statement. Unfortunately, the anatomical knowledge of the mantled baboon is still debatable. For sure, there is a large body of publications that elaborate on a very specific aspect of the baboon’s anatomy. A quick literature search on the Internet demonstrates the scattered availability of knowledge. However, those publications fall short when systematic, comprehensive information on an anatomical system of the baboon is sought after. A reference work worthwhile mentioning is ‘An Atlas of Primate Gross Anatomy: Baboon, Chimpanzee and Man’ [6]. Since the intention of this atlas is to clarify the anatomical relationships among the mentioned species, the anatomical descriptions of each species stay rather superfluous. This prompted the authors of the present study a few years ago to begin revisiting the anatomy of the mantled baboon. A first manuscript presenting an in-depth description of the osteology was published in this journal in 2023 [7]. As is custom in anatomical atlases and handbooks, the study of the joints (arthrology) follows. Surprisingly, no chapter on the arthrology of the baboon is included in the above-cited atlas. A similar finding is found when browsing through the renowned source of data on primate anatomy published by Ankel-Simons [8]. That work is further characterized by a comparative approach rather than providing species-specific details.

The present arthrological work focuses on the synovial joints of the appendicular skeleton and, more specifically, those of the thoracic limb (*articulationes membri thoracici*) of the mantled baboon. It is the first part of three, with the arthrology of the pelvic limb and the axial skeleton forthcoming. In contrast to humans, the baboon’s thoracic limb (arm) plays an important role in locomotion [9]. It should, consequently, withstand great forces, resulting in mechanical stresses on the bones, muscles, and joints. In addition, the primate hand is also used for precision handling [10]. These conflicting functions require that the size, stability, and sturdiness of the hand should be balanced with minimal weight, flexibility, and agility [11]. As such, knowledge of human anatomy and, in particular, biomechanics cannot be extrapolated to the mantled baboon. Nonetheless, veterinarians responsible for the medical care of baboons often rely on human anatomical works when wounds have to be tended or surgical interventions are required. The reason can be found in the fact that primates, including baboons and humans, share massive amounts of anatomical traits due to their close genetic relationship [12].

To facilitate the use of this manuscript as a dissection guide and reference work for veterinarians who are responsible for the medical care of baboons, all the synovial joints that are present in the baboon’s thoracic limb (i.e., those present in the shoulder region, elbow region, and hand) are topographically approached. This means that not only the typical connective-tissue elements that form the joints but also the associated muscle tendons are depicted by means of labeled color photographs of dissections. The osseous structures to which these components attach are identified as well. Veterinary anatomical terminology is primarily used and derived from the *Nomina Anatomica Veterinaria* (N.A.V.) [13]. Nevertheless, it is complemented by human terminology listed in the *Terminologia Anatomica* (T.A.) [14]. This approach has its foundation in the fact that only domestic mammals are included in the N.A.V. Structures specific for primates are not listed there but might be found in the T.A. considering the great similarity between humans and other primates [12].

The aim of the present work is to provide a thorough description of the gross anatomy of the thoracic limb joints of the mantled baboon, using contemporary anatomical terminology. The textual descriptions are supported by 25, often multipanel, color photographs, as a picture is worth a thousand words. Several points of view are presented from the superficial layer to the deepest, guiding the reader through the dissections. As such, the present work could be valuable as a dissection guide. It is not the intention of the work to provide a major comparative functional or evolutionary perspective. Though, the discussion will bring forward a few remarkable differences between the anatomy of the baboon, on the one hand, and humans and domestic mammals, on the other hand. This is to illustrate that the extrapolation of species-specific anatomical data should be performed with caution. Furthermore, providing functional and evolutionary insights also falls beyond the scope of the present work because too many structures are reviewed. This work could perhaps be a source of inspiration for further research on a particular matter. A potential research topic could be the radiographic examination of the baboon’s joints, as imaging exams are commonly used in the clinical routine for baboons. Such a complementary method for visualizing bone structures and joints could build further on the recently published manuscript on the osteology and the present manuscript on the arthrology of the mantled baboon.

## 2. Materials and Methods

### 2.1. Animals

The frozen cadavers of three adult hamadryas baboons (*Papio hamadryas*), one male and two females, were obtained from the Biobank of the Biomedical Primate Research Centre (BPRC), Rijswijk, The Netherlands (https://www.bprc.nl/en/biobanks/biobank, last accessed 28 June 2025). The involved animals were euthanized for several reasons, including welfare issues, population control, and research/diagnostics. These reasons were not related to the locomotor system and, therefore, did not influence the described morphology. All animals were fasted overnight and sedated by intramuscular injection of ketamine hydrochloride (10 mg/kg ketamine 10%^®^, Alfasan Diergeneesmiddelen B.V., Woerden, The Netherlands) combined with medetomidine hydrochloride (0.05 mg/kg, Sedastart^®^, AST Farma B.V., Oudewater, The Netherlands). They were subsequently euthanized with an intravenous (*v. saphena parva*) pentobarbital dose (60 mg/kg) (Euthasol^®^ 20%; AST Farma B.V.). It is described that this pharmacological compound does not induce tissue damage, not even at the histopathological level [15]. The frozen cadavers were transported to the Faculty of Veterinary Medicine, Ghent University, Belgium, where they were stored at −20 °C. The anatomical examinations of the thoracic limbs were initiated once the cadavers had been thawed sufficiently at room temperature.

### 2.2. Dissection Procedure

Both the left and right thoracic limbs of the three mantled baboons were examined. However, only photographs of the left thoracic limbs are presented, since the musculoskeletal system is traditionally depicted on the left limb in most anatomy books and atlases. First, the skin and superficial fasciae overlying the thoracic limbs, including the shoulder region, and the thoracic walls were removed. To this purpose, a first incision was made along the dorsal midline from the occiput to the iliac crest (*crista iliaca*). A second incision was made along the ventral midline from the groin to the pecten pubis. In addition, a longitudinal incision was made in proximodistal direction at the lateral side of the thoracic limb from the dorsal midline to the digits. An analogous incision was made at the medial side, starting at the ventral midline. All incisions followed the direction of the hair and skin tension lines. Removing the skin started from these incisions working toward each other. The superficial fascia was subsequently removed, and the deep fascia was resected along the borders of the muscles, revealing the delineation between the several muscles. After depicting the junctional muscles that attach the thoracic limb to the thorax, they were transversely cut. Next, the sternum was transversely sawn at the level of the second intercostal space, and the first two ribs were bisected by means of an oscillating saw. As a result, the thoracic limbs were isolated as study objects, while the joints between the clavicle (*clavicula*), on the one hand, and the sternum and scapula, on the other hand, remained intact.

For each joint, except for those of the hand that was studied in its entirety, the adjoining muscles with their tendons were first studied. In this respect, the work by Diogo and Wood on the comparative anatomy of primate muscles was very valuable, as it describes and shows the musculature in a large number of primate species, including *Papio* [16]. The Visible Ape Project was another very helpful tool that was used for gaining insight into the organization of the various muscles in each of the studied regions [17]. Subsequently, the joints were isolated by bisecting the neighboring bones at the midpoint of their diaphyses by means of an oscillating saw. Then, all studied muscles were resected, except for their tendinous origins and/or insertions. The excess of soft tissues was macerated using a brush dipped in sodium hypochlorite 42% (Brunschwig Chemie, Amsterdam, The Netherlands). Respective volumes of 8 mL, 6 mL, and 1 mL polymethyl methacrylate (PMMA) (Batson’s #17, Polysciences Europe GmbH, Hirschberg an der Bergstrasse, Germany) were injected into the synovial joint cavities of the shoulder, elbow, and sternoclavicular joints of one female mantled baboon. All obtained joints were submerged overnight in tap-water-diluted hydrogen peroxide 27.5% (Brunschwig Chemie, Amsterdam, The Netherlands) for bleaching purposes. Then, the specimens were rinsed with tap water and dried at room temperature. Finally, they were degreased by means of methylene chloride (dichloromethane) (Brunschwig Chemie, Amsterdam, The Netherlands) through a distillation process (hand-built apparatus).

### 2.3. Data Presentation

Color photographs of the specimens were taken during the subsequent dissection steps, from the superficial to the deeper layers, from several points of view. These were often arranged as multipanel figures, allowing for the use of the manuscript as a dissection guide. The orientation of the specimens are indicated on the images using the following abbreviations: Ca = caudal, Cr = cranial, D = distal, Do = dorsal, L = lateral, M = medial, P = proximal, Pa = palmar, and Ve = ventral. A Canon EOS M50 body (Canon Inc., Tokyo, Japan) equipped with a Canon EF-M 18–150 mm f/3.5–6.3 IS STM lens (Canon Inc., Tokyo, Japan) was operated. Cropping, color correcting, enhancing contrast and brightness, and harmonizing the black background were performed using GIMP 2.10.38. (gimp.org, last accessed 21 July 2025).

The N.A.V. [13] and T.A. [14] were consulted to label the structures in PowerPoint (Microsoft, Redmond, WA,USA). English terminology is consistently used in the text. However, the first time a term appears in the text, the Latin term is provided in italics between brackets. In contrast, Latin terms are exclusively used in figure legends.

## 3. Results

### 3.1. Introduction

In this section, the various synovial joints of the thoracic limb are discussed from the proximal to the distal regions (*regiones membri thoracici*), thus first those situated in the shoulder region (*regio articulationis humeri*), then those in the elbow region (*regio cubiti*), and ending with the joints present at the level of the carpus (*regio carpi*) and the numerous joints located in the hand (*regio metacarpi*,* regio metacarpophalangea*,* regio phalangis proximalis*,* regio interphalangea proximalis*, and *regio phalangis mediae*). Due to the presence of the clavicle, two additional joints that are absent in any of the domestic mammals are located in the region of the scapula (*regio scapularis*), more specifically where the acromion is situated (*regio acromialis*), and in the region of the sternum (*regio sternalis*). The general conformation of each region is portrayed first. The muscles surrounding each joint are subsequently described, from the superficial to the deeper layers. Finally, the arthrological structures assembling each joint are elucidated, comparing pre- with post-maceration photographs.

### 3.2. Shoulder Region and Acromial Region

#### 3.2.1. General Conformation

The three bones composing the shoulder or pectoral girdle are the shoulder blade or scapula, humerus, and collar bone, or clavicle. The articulation present in the shoulder region is the shoulder joint (*articulatio *(*art.*)* glenohumeralis* syn. *art. humeri*). It is formed between the concave glenoid cavity of the scapula (*cavitas glenoidalis*) and the convex head of the humerus (*caput humeri*). Within the acromial region, the acromioclavicular joint (*art. acromioclavicularis*) that connects the acromion of the scapula with the acromial extremity of the clavicle (*extremitas acromialis*) can be found. Because both joints are in close proximity, they are discussed simultaneously.

Since muscles form a significant reinforcement of the joints that are typically composed of the joint capsule (*capsula articularis*) and various ligamentous structures, they are described first in the paragraph entitled ‘Myology’. Moreover, they lie superficial to the joints and are encountered first during dissection. The components belonging to the joints themselves are subsequently portrayed in the paragraph entitled ‘Arthrology’.

#### 3.2.2. Myology

##### Lateral and Dorsal Views

A few superficial muscles are easily recognized after removing the skin at the lateral and dorsal aspects of the shoulder region (Figure 1A–C). The trapezius muscle (*musculus *(*m.*)* trapezius*), which originates at the dorsal midline of the neck (*pars cervicalis*) (Figure 1A and Figure 2A, no. 1a) and back (*pars thoracica*) (Figure 1A and Figure 2A, no. 1b) inserts primarily on the scapular spine. The cervical part attaches to the entire length of the scapular spine, including the acromion, as well as to the acromial extremity of the clavicle, whereas the thoracic part only attaches to the proximal third of the scapular spine. Caudal to this part lies the latissimus dorsi muscle (*m. latissimus dorsi*) (Figure 1A,B, no. 2), which originates by means of an aponeurosis along the dorsal midline, at the level of the sixth to twelfth thoracic vertebrae, and the lumbodorsal fascia. The insertion site is the tuberosity of the teres major muscle (*tuberositas teres major*), where the teres major muscle (*m. teres major*) also attaches. The deltoid muscle (*m. deltoideus*) consists of three parts. The clavicular part (*pars clavicularis*) (Figure 2A, no. 2a) starts on the lateral half of the clavicle (Figure 2A, C), the acromial part (*pars acromialis*) (Figure 1A, no. 3a, and Figure 2A, no. 2b) originates on the acromion of the scapula (Figure 2A, S), and the scapular spinal part (*pars spinalis scapularis*) (Figure 1A, no. 3b, and Figure 2A, no. 2c) has its origin along the entire length of the scapular spine (*spina scapulae*). The three parts distally unite to insert into the deltoid tuberosity (*tuberositas deltoidea*) of the humerus (Figure 2A, H). The dorsal view allows for the identification of the trapezius muscle (Figure 1B, nos. 1a and 1b) and the latissimus dorsi muscle with its aponeurosis (Figure 1B, no. 2), which both meet in the dorsal midline.

After removing these superficial muscles, the deeper muscle layer becomes visible (Figure 2B). In the dorsal view, the rhomboideus muscle (*m. rhomboideus*) and the levator scapulae (*m. levator scapulae*) muscle can be seen. The former muscle is composed of a thoracic part (*m. rhomboideus thoracis*) (Figure 1C, no. 6a) attaching to the spinal processes of the fourth to sixth thoracic vertebrae, a cervical part (*m. rhomboideus cervicis*) (Figure 1C, no. 6b) attaching to the spinal processes of the cranially located vertebrae, and a part attaching to the occiput (*m. rhomboideus capitis*) (Figure 1C, no. 6b). They all start from the medial side of the dorsal margin of the scapula. The levator scapulae muscle can be found ventral to the rhomboideus capitis muscle. It runs from the cranial angle of the scapula to the atlas. The following muscles can be identified in the lateral view. The omotransversarius muscle (*m. omotransversarius*) (also known as *m. levator claviculae*) (Figure 2B, no. 3) is revealed after resection of the cervical part of the trapezius muscle. It starts on the transverse processes (*processus transversi*, singular: *processus transversus*) of the first and second cervical vertebrae (*atlas* or C1 and *axis* or C2, respectively), runs medial (deep) to the cervical part of the trapezius muscle to insert into the acromion. As such, it overlies the acromioclavicular joint. Additionally, the supraspinatus muscle (*m. supraspinatus*) (Figure 2B, no. 4) becomes discernable. This muscle originates in the supraspinous fossa (*fossa supraspinata*) that is located cranial to the scapular spine and inserts into the greater tubercle of the humerus (*tuberculum majus*). Resection of the clavicular part of the deltoid muscles shows the insertion of the subclavius muscle (*m. subclavius*) (Figure 2B, no. 5) into its groove on the clavicle (*sulcus musculi subclavii*). The origin of this muscle is the cartilage of the first rib. Removing the acromial part of the deltoid muscle exposes the biceps brachii muscle (*m. biceps brachii*) (Figure 2B, no. 6), which will be elaborated further. Finally, after the resection of the scapular part of the deltoid muscle, the infraspinatus muscle (*m. infraspinatus*) (Figure 1A, no. 4, and Figure 2B, no. 7) can be examined. It is located caudal to scapular spine, in the infraspinous fossa (*fossa infraspinata*), and inserts immediately caudal to the supraspinatus muscle into the smooth surface just distal to the laterally projected caudal aspect of the greater tubercle (*facies musculi infraspinati*).

During manipulation, it could be noticed that the tendons of the supra- and infraspinatus muscles stabilize the shoulder joint by laterally embracing the humeral head. Also, the teres minor muscle (*m. teres minor*) (Figure 2B, no. 8) is now fully exposed. It begins on the distal third of the caudal border of the scapula (*margo caudalis scapulae*) and runs caudal to the infraspinatus muscle toward the tuberosity of the teres minor muscle (*tuberositas teres minor*), immediately distal to the greater tubercle of the humerus. A minor shoulder stabilizing effect could be perceived during manipulation. Furthermore, two heads of the triceps brachii muscle (*m. triceps brachii*) (i.e., the lateral head (*caput laterale musculi tricipitis brachii*) (Figure 2B, no. 9) and the long head (*caput longum musculi tricipitis brachii*) (Figure 1A, no. 5, and Figure 2B, no. 10)) are in the field of view. The former starts at the tricipital line (*linea musculi tricipitis*) (i.e., the distal twig of the caudal portion of the greater tubercle of the humerus), while the latter originates at the caudal border (*margo caudalis*) and the infraglenoid tubercle (*tuberculum infraglenoidale*) of the scapula. Both heads course toward the olecranon tuber (*tuber olecrani*) of the ulna. The long head covers the latissimus dorsi muscle (Figure 2B, no. 11) laterally.

##### Medial View

The serratus ventralis muscle (*m. serratus ventralis*) attaches the shoulder blade and, by extension, the shoulder girdle to the thoracic cage (*m. serratus ventralis thoracis*) (Figure 3A, no. 1) and to the cervical part of the vertebral column (*m. serratus ventralis cervicis*) (Figure 3A, no. 2). This muscle had to be cut transversely when isolating the thoracic limb for examination. The subclavius muscle (Figure 3A, no. 3) can be observed attaching to the clavicle. The teres major muscle (Figure 3A, no. 4) was touched upon during the description of the lateral side. Its insertion together with the latissimus dorsi muscle (Figure 3A, no. 5) has been defined there. Its origin can be found immediately ventral to the caudal angle of the scapula (*angulus caudalis scapulae*), where the caudal margin of the scapula (*margo caudalis scapulae*) is broader. The subscapularis muscle (*m. subscapularis*) fills the entire subscapular fossa (*fossa subscapularis*) (Figure 3A–F, no. 6). Its tendon terminates on the lesser tubercle (*tuberculum minus*) of the humerus, thus traversing the shoulder joint medially in proximo-distal direction. Its tendon is, however, obscured by a number of medial muscles of the upper arm. The coracobrachialis muscle (*m. coracobrachialis*) also crosses the shoulder joint, albeit in the caudodistal direction, from the coracoid process to the humerus. The cranially located, deeper coracobrachialis muscle (*m. coracobrachialis profundus*) (Figure 3A,C,D, no. 7a) inserts into the humeral neck (*collum humeri*), while the caudally located, middle coracobrachialis muscle (*m. coracobrachialis medius*) (Figure 3A–D, no. 7b) ends more distally at the medial side of the humeral shaft. The short head of the biceps brachii muscle (*caput breve musculi bicipitis brachii*) (Figure 3A–C, no. 8a) that also begins on the coracoid process travels in between both coracobrachialis muscles. In contrast, the long head of the biceps brachii muscle (*caput longum musculi bicipitis brachii*) (Figure 3A–E, no. 8b) finds its origin at the supraglenoid tubercle (*tuberculum supraglenoidale*) of the scapula, travels through the intertubercular groove and terminates on the radial tuberosity (*tuberositas radii*) and the interosseous membrane, together with the short head.

The acromioclavicular joint can be visualized by detaching the subclavius muscle from the clavicle (Figure 3B–F, no. 9). In addition, the coracoclavicular ligament (Figure 3B–F, no. 10) becomes discernable. To reach the medial side of the articular capsule of the shoulder joint, the subscapularis muscle must be detached from its origin in the subscapular fossa and retracted toward its insertion into the lesser humeral tubercle. To this purpose, the medial muscles of the upper arm were systematically removed. The teres major and latissimus dorsi muscles were cut close to their common insertion so that only a stump remained (Figure 3B–F, no. 11). The short head of the biceps brachii muscle was first transversely cut halfway the humerus and retracted toward its origin on the coracoid process (Figure 3C, no. 8a). It was ultimately removed. This allowed for the better visualization of the deep and middle coracobrachialis muscles (Figure 3C,D, no. 7a and 7b, respectively). After their resection, the tendon of the subscapularis muscle was fully exposed (Figure 3E, no. 12). Finally, detaching the subscapularis muscle and distally retracting it revealed the medial side of the capsule of the shoulder joint (Figure 3F, no. 13).

#### 3.2.3. Arthrology

##### Articulatio Humeri

The articular capsule of the shoulder joint (Figure 4A–C, no. 1) can be studied once all the obscuring muscles and tendons have been resected. It is spanned between the margins of the scapular glenoid cavity and the humeral neck. Gentle maceration of the soft tissues consumes the articular capsule, with the exception of the glenohumeral *ligaments *(*ligamenta *(*ligg.*)* glenohumeralia*), which are reinforcements of the joint capsule persisting to the maceration for longer time. A cranial glenohumeral ligament (*lig. glenohumerale craniale*) (Figure 4B no. 2) and a caudal glenohumeral ligament (*lig. glenohumerale caudale*), (Figure 4B,D no. 3) can be distinguished. They are located in the cranial and caudal walls of the joint capsule, respectively. Where the joint capsule between the glenohumeral ligaments is fully macerated, the articular cavity (*cavitas articularis*) (Figure 4B,D, no. 4) is accessible.

No collateral ligaments stabilizing the joint were observed. However, it was detected during the dissection that the stability is ensured by the tendons of the above-described supraspinatus, infraspinatus, teres minor, and subscapularis muscles, also known as the rotator cuff muscles in human medicine [11]. The inserting tendons of these respective muscles are labeled in Figure 4B (nos. 5, 6, and 7) and Figure 4D (no. 8). Dissecting the tendon of the infraspinatus muscle that runs adjacent to the lateral wall of the articular capsule demonstrates the presence of a subtendinous bursa (*bursa musculi infraspinati*) (Figure 4B, no. 9) in between the tendon and the greater humeral tubercle. The insertion of the long head of the triceps brachii muscle is additionally indicated in Figure 3B,D (no. 10). In Figure 4D, the tendon of origin of the biceps brachii muscle (no. 11), which will be further elaborated, is indicated as well.

The coracohumeral ligament (*lig. coracohumerale*) (Figure 4B,D, no. 12) presents an oblique course along the cranial side of the articular capsule, from the caudally oriented tip of the scapular coracoid process medially to the greater humeral tubercle laterally. It strengthens the cranial aspect of the joint capsule with which it blends. The coracoclavicular ligament (*lig. coracoclaviculare*) (Figure 4B,D, no. 13) and the acromioclavicular ligament (*lig. acromioclaviculare*) (Figure 4B, no. 14) is discussed below when describing the acromioclavicular joint.

Exploring the articular cavity shows that the spheroidal glenohumeral joint provides the connection between the socket-like glenoid fossa of the scapula and the ball-like head of the humerus. The shallow glenoid fossa (Figure 5A, no. 1) is expanded by the glenoid labrum (*labrum glenoidale*) (Figure 5A, no. 2), which is a rim of fibrous cartilage fitting onto its osseous margins. The articular capsule (Figure 5A, no. 3) originates from the exterior surface of this labrum. In between the articular capsule and the glenoid labrum originates the long head of the biceps brachii muscle on the supraglenoid tubercle (Figure 5A, no. 4). The insertion of the joint capsule (Figure 5B, no. 5) on the humerus is at its neck. This is well understood when examining a shoulder joint of which the joint cavity has been injected with polymethyl methacrylate (Figure 5B, no. 6). Partially dissolving the polymethyl methacrylate injectate by means of methylene chloride (Figure 5C, no. 7) reveals that the greater and lesser tubercles are not enclosed by the joint capsule. Indeed, the injectate is lodged at the border between the greater tubercle (Figure 5C, no. 8) and the humeral head (Figure 5C, no. 9), thus at the anatomical neck (Figure 5C, no. 10). At the caudal side of the joint, the joint capsule bulges in the distal direction forming the axillary recess (*recessus axillaris*) (Figure 5C, no. 11).

The tendon of the long head of the biceps brachii muscle (Figure 6A,B, no. 1) runs in the articular cavity, since it attaches to the supraglenoid tubercle in between the articular capsule and the glenoid labrum. This tendon is surrounded by the articular capsule where it runs in the intertubercular groove. The extension of the articular capsule surrounding this tendon is called the intertubercular tendon sheath (*vagina tendinis intertubercularis*). When injecting the articular cavity with polymethyl methacrylate, the injectate fills the tendon sheath (Figure 6A,B, no. 2). The transverse humeral ligament (*ligamentum transversum humeri*) (Figure 6A,B, no. 3), or *retinaculum transversum* according to the N.A.V. [9], secures the tendon of the biceps brachii muscle in the intertubercular groove. It is stretched between the distal aspects of the lesser and greater tubercles of the humerus (Figure 6A,B, nos. 4 and 5, respectively). In Figure 6A,B, the insertions of the m. supraspinatus (no. 6) and m. infraspinatus (no. 7) into the greater tubercle are additionally indicated. The m. subscapularis inserts into the lesser tubercle (Figure 6A,B, no. 8). The coracohumeral ligament (Figure 6A,B, no. 9), extending between the coracoid process and the greater tubercle, and the coracoclavicular ligament (Figure 6A,B, no. 10), extending between the coracoid process and the clavicle, are well visible in the cranial view of the shoulder joint.

##### Articulatio Acromioclavicularis

The acromioclavicular joint is a synovial joint formed by the acromion of the scapula (Figure 7A–C, no. 1) and the acromial extremity of the clavicle (Figure 7A–C, no. 2). In between both sits the fibrocartilaginous articular disc of the acromioclavicular joint (*discus articularis acromioclavicularis*) (Figure 7A, no. 3). This synovial joint presents a joint cavity, surrounded by a joint capsule (Figure 7A,B, no. 4), which can be injected with polymethyl methacrylate (Figure 7C, no. 5). Craniodorsally, the joint capsule is reinforced by the acromioclavicular ligament (Figure 7C, no. 6), which extends between the dorsolateral aspect of the acromial extremity of the clavicle and the craniodorsal margin of the acromion. It, thus, covers the acromioclavicular joint craniodorsally.

### 3.3. Sternal Region

#### 3.3.1. General Conformation

The joint present in the sternal region that provides the connection between the sternum and the clavicle is the sternoclavicular joint (*art. sternoclavicularis*). It is the joint that connects the thoracic limb and, more specifically, the shoulder girdle with the thorax.

#### 3.3.2. Myology

A number of muscles attach to the surrounding of the sternoclavicular joint, partially obscuring it. The sternocleidomastoideus muscle (*m. sternocleidomastoideus*) is a tripartite muscle. It consists of the medial sternomastoideus muscle (*m. sternomastoideus*) (Figure 8A, no. 1) that runs from the manubrium of the sternum (*manubrium sterni*) to the mastoid process of the skull, the middle cleidomastoideus muscle (*m. cleidomastoideus*) (Figure 8A, no. 2) that starts from the sternoclavicular joint and also runs to the mastoid process, and the lateral cleidooccipitalis muscle (*m. cleidooccipitalis*) (Figure 8A, no. 3) that arises from sternal side of the clavicle and inserts into the nuchal line of the skull. At the cranial tip of the manubrium arises the sternohyoideus muscle (*m. sternohyoideus*) (Figure 8B, no. 4) that inserts into the medial aspect of the hyoid bone (*os hyoideum*). Immediately lateral to it runs the sternothyroideus muscle (*m. sternothyroideus*) (Figure 8B, no. 5) to the thyroid cartilage (*cartilago thyroidea*). The superficial pectoral muscle (*m. pectoralis superficialis*) consists of three parts. The smaller, cranially located sternocapsular part (*pars sternocapsularis*) (Figure 8C, no. 6) is initiated caudal to the sternohyoideus muscle on the manubrium of the sternum and the medial aspect of the sternoclavicular joint and attaches to the proximal area of the deltoid tuberosity, immediately distal to the insertion of the clavicular part of the deltoid muscle (Figure 8C,D, no. 7). Resection of the above-mentioned muscles fully exposes the sternoclavicular joint (Figure 8D). The very broad, middle sternal part (*pars sternalis*) of the superficial pectoral muscle (Figure 8D,E, no. 8) that lies caudal to the sternocapsular part starts from the entire length of the sternum and attaches to the distal aspect of the deltoid tuberosity. Along its caudolateral border, the caudally located abdominal part (*pars abdominalis*) (Figure 8D–H, no. 9) is visible. It courses from the costal arch (*arcus costalis*) adjacent to the xiphoid process and the cranial aspect of the rectus abdominis sheath to the lesser humeral tubercle. Along its course in cranial direction, it becomes covered by the sternal part. The removal of the clavicular part of the deltoid muscles allows for the inspection of the subclavius muscle (Figure 8E, no. 10). This pectoral muscle was already described above and depicted in Figure 1B. The deep pectoral muscle (*m. pectoralis profundus*) (Figure 8E–H, no. 11) lies deep to the sternocapsular and sternal parts of the superficial pectoral muscle. It has its origin on the body of the sternum (*corpus sterni*) and the cartilages of the second to sixth ribs. It travels transversely to the longitudinal axis of the body toward the greater tuberosity of the humerus, meeting the abdominal part of the superficial pectoral muscle at its caudal border. However, the insertion is covered by the tendon of the supraspinatus muscle (Figure 8E,F, no. 12). In Figure 8F, the supraspinatus muscle (no. 12) and the infraspinatus muscle (no. 13) were detached from their respective fossae and retracted toward their insertions. The resection of the subclavius muscle reveals the coracoclavicular ligament (Figure 8E–H, no. 14) at its lateral border. As a result, the tendon of the supraspinatus muscle travels through a tunnel enclosed by the clavicle cranially, the articular capsule of the shoulder joint caudally, the acromion laterally, and the coracoclavicular ligament medially. Removing the stumps of the supraspinatus and infraspinatus muscles demonstrates that the tendon of the deep pectoral muscle crosses the intertubercular groove (*sulcus intertubercularis*) to attach to the greater tuberosity (Figure 8G, no. 15). The long head of the biceps brachii muscle (Figure 8H, no. 16) travels in this groove. The short head runs medial to the long head of the biceps brachii muscle (Figure 8H, no. 17).

#### 3.3.3. Arthrology

The ball of the sternal extremity of the clavicle (*extremitas sternalis*) (Figure 9A,B, no. 1) and the socket at the craniolateral side of the manubrium (clavicular notch, *incisura clavicularis*) (Figure 9A,B, no. 2) form the sternoclavicular joint. In Figure 9A, the articular cavity is casted by means of polymethyl methacrylate (no. 3). An articulating disc (*discus articulationis sternoclavicularis*) sits in the articular cavity (insert in Figure 9A), which is delimited by the articular capsule (Figure 9B, no. 4). The reinforcements of the articular capsule that resist the soft-tissue maceration for a longer time are the ventral and dorsal sternoclavicular ligaments (*lig. sternoclaviculare ventrale* and *dorsale*, respectively) (Figure 9A, no. 5, and Figure 9B, no. 6, respectively). The interclavicular ligament (*lig. interclaviculare*) (Figure 9A,B, no. 7) is a thin ligament that connects the cranial margins of the sternal extremities of the left and right clavicles. To this purpose, the ligament travels along the cranial border of the sternal manubrium.

### 3.4. Elbow Region

#### 3.4.1. General Conformation

The skeletal elements present in the elbow region are the humerus, radius, and ulna. Each of the three bones forms a joint with the other two. Consequently, three synovial joints are located within the elbow region, i.e., humeroulnar joint (*art. humeroulnaris*), humeroradial joint (*art. humeroradialis*), and proximal radioulnar joint (*art. radioulnaris proximalis*). The former two are assembled in the elbow joint (*art. cubiti*). However, all three individual articulations are included in a single synovial-lined joint capsule.

The elbow joint allows for movement in the sagittal plane, i.e., flexion and extension of the elbow. The humeroulnar joint is a hinge joint in which the ulnar trochlear notch (*incisura trochlearis*) articulates with the trochlea of the humerus (*trochlea humeri*). In contrast, the humeroradial joint is formed between the convex capitulum of the humerus (*capitulum humeri*) and the concave articular facet (*fovea articularis capitis radii*) of the radial head (*caput radii*).

The proximal radioulnar joint is a pivot joint that permits the radius and ulna to revolve along the transversal plane so that the forearm exhibits a rotational movement, both internally and externally, called pronation and supination, respectively. Here, the articular circumference (*circumferentia articularis*) of the radial head connects with the radial notch (*incisura radialis*) of the ulna, which is located lateral and somewhat distal to the coronoid process (*processus coronoideus*).

Multiple muscles find their origins in the elbow region. These include, among a few others, the extensor and flexor muscles of the carpus and digits. A smaller number of muscles, such as the biceps brachii and triceps brachii muscles, have their insertions in this region. The musculature is described first in the paragraph entitled ‘Myology’. The various ligaments that preserve the integrity of the joint during movement and utilization are subsequently discussed in the paragraph entitled ‘Arthrology’.

#### 3.4.2. Myology

##### Lateral View

Four muscles are located at the caudal side of the brachium and can be dissected from the lateral side. The lateral head (*caput laterale*) (Figure 10, no. 1) and the long head (Figure 10, no. 2) of the triceps brachii muscle insert jointly by a common tendon into the olecranon tuber. Nevertheless, their origins differ significantly. The lateral head of the triceps muscle originates from the tricipital line, while the long head has its origin at the infraglenoid tubercle, and was relevant in the discussion of the shoulder joint. The dorsoepitrochlearis muscle (*m. dorsoepitrochlearis*) (Figure 10, no. 3) arises from the ventral margin of the latissimus dorsi muscle (*m. latissimus dorsi*). It merges by means of a broad aponeurosis with the long head of the triceps brachii muscle and, finally, inserts jointly on the olecranon tuber. The anconeus muscle (*m. anconeus*) (Figure 10, no. 4) arises from the caudal side of the distal third of the humerus and attaches to the anconeal process (*processus anconeus*) of the ulna. The inserting muscle fibers are intimately associated with the dorsal side of the articular capsule. The here-described muscles extend the elbow.

The brachioradialis muscle (*m. brachioradialis)* (Figure 10, no. 5) is the only flexor muscle of the elbow that is discussed by means of the lateral view. It runs from the proximal half of the lateral supracondylar ridge of the humerus (*crista supracondylaris lateralis* syn. *crista supinatoris*) to the lateral side of the styloid process of the radius (*processus styloideus radii*). The insertions of the biceps brachii muscle (Figure 10, no. 6) and the brachialis muscle (Figure 10, no. 7) are obscured by the origin of the extensor carpi radialis muscle (*m. extensor carpi radialis*) (Figure 10, no. 8a and 8b).

The extensor carpi radialis muscle comprises two heads. The extensor carpi radialis longus muscle (*m. extensor carpi radialis longus*) (Figure 10, no. 8a) has its origin at the lateral supracondylar ridge of the humerus, distal to the origin of the brachioradialis muscle, while the extensor carpi radialis brevis muscle (*m. extensor carpi radialis brevis*) (Figure 10, no. 8b) commences on the lateral humeral epicondyle (*epicondylus humeri lateralis*). Several extensor muscles of the carpus and digits arise here. The tendon of origin of the more caudally located extensor digitorum communis muscle (*m. extensor digitorum communis*) (Figure 10, no. 9) is merged with the lateral collateral ligament of the elbow joint (*lig. collaterale cubiti laterale*). The tendons of origin of the extensor muscle for the fourth digit (*m. extensor digiti quarti*) (Figure 10, no. 10) and the fifth digit (*m. extensor digiti quinti/minimi*) (Figure 10, no. 11) can be found just caudal to the former. Finally, the extensor carpi ulnaris muscle (*m. extensor carpi ulnaris*) (Figure 10, no. 12) commences most caudally on the lateral humeral epicondyle. The extensor pollicis longus muscle (*m. extensor digiti primi/pollicis longus*) (Figure 10, no. 13) becomes apparent distally, in between the common extensor muscle and the extensor muscle of the fourth digit. Nevertheless, it is irrelevant here as its origin is craniolaterally on the proximal half of the ulna. Cranial to it lies the abductor pollicis longus muscle (*m. abductor digiti primi/pollicis longus*) (Figure 10, no. 14) that arises from the proximolateral aspect of the ulna and the cranial side of the radius. This muscle is also unrelated to the elbow joint. The flexor carpi ulnaris muscle (*m. flexor carpi ulnaris*) (Figure 10, no. 15) is located caudalmost on the antebrachium. As it has its origin on the medial side of the elbow joint, it is discussed below using the medial view.

##### Medial View

The flexor muscles of the carpus and digits have their origins at the medial side of the elbow region. The flexor carpi ulnaris muscle that was already visible from the lateral side is easily recognized in the medial view, as it lies most caudally in the antebrachium. Its caudally positioned ulnar head (*caput ulnare*) (Figure 11A,B, no. 1a) initiates on the olecranon and the concavity caudal to the trochlear notch, whereas its cranially positioned humeral head (*caput humerale*) (Figure 11A,B, no. 1b) starts at the caudal aspect of the medial humeral epicondyle (*epicondylus medialis humeri*). The palmaris longus muscle (*m. palmaris longus*) (Figure 11A,B, no. 2) is situated immediately cranial to the flexor carpi ulnaris muscles. It runs from the medial humeral epicondyle to the palmar fascia (*fascia palmaris*). Cranial to this muscle and also initiated at the medial humeral epicondyle sits the flexor carpi radialis muscle (*m. flexor carpi radialis*) (Figure 11A,B, no. 3). Its muscle belly is cranially flanked by the pronator teres muscle (*m. pronator teres*) (Figure 11A,B, no. 4), which also starts at the medial humeral epicondyle and attaches at the midpoint of the lateral radial diaphysis. This muscle is, however, no flexor of the carpus or digits but endorotates the antebrachium. The deeper layer consists of the caudally located flexor digitorum superficialis muscle (*m. flexor digitorum superficialis*) (Figure 11A,B, no. 5) and the more cranially located flexor digitorum profundus muscle (*m. flexor digitorum profundus*) (Figure 11A, no. 6). The superficial flexor muscle of the digits consists of a single muscle belly with its origin on the medial humeral epicondyle. In contrast, the deep flexor muscle of the digits has two heads in addition to the head that has its origin on the medial humeral epicondyle (*caput humerale*) (Figure 11B, no. 6a). These are the ulnar head (*caput ulnare*) that arises from the proximal half of the ulna, just proximal to the ulnar tuberosity (*tuberositas ulnae*) (Figure 11B, no. 6b), and the radial head, which originates at the upper two-thirds of the radius (Figure 11B, no. 6c).

Resecting the flexor muscles of the carpus and digits and the pronator teres muscle, allows for the study of the flexor musculature of the elbow joint. The brachioradialis muscle (Figure 11A–C, no. 7) has been discussed above using the lateral view. It is, however, also visible in the medial view, since it is the most cranially located muscle of the antebrachium. Immediately caudal to it, also evident in the medial view, lies the extensor carpi radialis longus muscle (Figure 11B,C, no. 8). The brachialis muscle (Figure 11A–C, no. 9) starts at the lateroproximal aspect of the humeral shaft, initially covers the lateral surface of the humeral body (*corpus humeri facies lateralis*) to twist to its cranial surface (*corpus humeri facies lateralis*), and, finally, inserts distal to the medially oriented coronoid process of the ulna into the ulnar tuberosity (*tuberositas ulnae*). The biceps brachii muscle (Figure 11A,B, no. 10) presents the medially positioned short head (Figure 11C, no. 10a) and the long head (Figure 11C, no. 10b) that is placed laterally. Both heads unite distally to insert with a single tendon into the radial tuberosity (*tuberositas radii*), while also attaching to the interosseous membrane. This insertion can be found immediately distal and lateral to the insertion of the brachialis muscle.

The extensor musculature of the elbow joint consists of the tensor fasciae antebrachii muscle, the triceps brachii muscle, the dorsoepithrochlearis muscle, the anconeus muscle and the epitrochleoanconeus muscle. The tensor fasciae antebrachii muscle (*m. tensor fasciae antebrachii*) (Figure 11A, no. 11) lies medial to the triceps brachii muscle, running from the caudoventral side of the latissimus dorsi muscle to the olecranon and antebrachial fascia (*fascia antebrachia*). The lateral head and the long head of the triceps brachii muscle (Figure 11A–C, no. 12) have already been described above using the lateral view. There, the dorsoepithrochlearis muscle is also mentioned. The medial head (*caput mediale*) of the triceps brachii muscle (Figure 11B,C, no. 13) arises from the medial side of the proximal third of the humeral shaft. The anconeus muscle (Figure 11A,C, no. 14) has also been described above. It can be observed craniomedial to the medial head of the triceps brachii muscle. Its muscle fibers initiate on the medial side of the middle and distal third of the humeral shaft. Finally, the epithrochleoanconeus muscle (*m. epithrochleoanconeus*) (Figure 11B insert, no. 15) is a short muscle that runs from the medial humeral epicondyle to the olecranon.

#### 3.4.3. Arthrology

Resecting the stumps of the muscles that have either their origins or insertions in the proximity of the elbow joint is likely to incise the articular capsule, since these structures are in close contact. The cranial portion of the articular capsule (Figure 12A,B, no. 1) encloses the entire humeral trochlea (Figure 12C, no. 2), the coronoid process (Figure 12C, no. 3), the radial notch of the ulna, and the head of the radius (Figure 12C, no. 4). Proximally, however, it also incorporates the coronoid fossa (*fossa coronoidea*) (Figure 12C, no. 5) medially and the radial fossa (*fossa radialis*) (Figure 12C, no. 6) laterally. Macerating the articular capsule reveals a number of ligaments. The head of the radius is constrained against the ulna by means of the annular ligament of the radius (*lig. anulare radii*) (Figure 12C, no. 7). It is attached to the ulnar coronoid process and the radial notch, thus embracing the radial head. Its function is to guide the rotational movement in the proximal radio-ulnar joint. This ligament merges with the lateral collateral ligament of the elbow joint (Figure 12C, no. 8), which is discussed below using the lateral view. In the cranial view, only the thin distal part (*pars distalis*) of the medial collateral ligament (*lig. collaterale cubiti mediale*) (Figure 12C, no. 9) can be observed. It is elaborated using the medial view, which will also be valuable for describing the insertions of the biceps brachii muscle (Figure 12A–C, no. 10) and the brachialis muscle (Figure 12A–C, no. 11).

The caudal view demonstrates that the articular capsule (Figure 13A,B, no. 1) attaches to the margins of the olecranon fossa (*fossa olecrani*) of the humerus (Figure 13C, no. 2) and encloses the trochlear notch of the ulna. The olecranon tuber (Figure 13A–C, no. 3) remains uncovered. Distally, the capsule merges with the lateral collateral ligament (Figure 13A–C, no. 4), which in turn is joined with the annular ligament of the radius (Figure 13A–C, no. 5). However, the lateral collateral ligament is not attached to the radial notch (Figure 13A–C, no. 6) but somewhat more distal. At the medial side, the joint capsule reaches the proximal part (*pars proximalis*) of the medial collateral ligament (Figure 13A–C, no. 7a). Indeed, the medial collateral ligament has a broad proximal and a thin distal part (Figure 13A–C, no. 7b). The proximal part attaches to the coronoid process (Figure 13A–C, no. 8), while the distal part inserts more distally. The tendon of the brachialis muscle (Figure 13B,C, no. 9) can be seen inserting in between both parts of the medial collateral ligament.

The lateral collateral ligament of the elbow joint (Figure 14A–C, no. 1) starts at the lateral epicondyle of the humerus (Figure 14A–C, no. 2), crosses the annular ligament of the radius (Figure 14, no. 3), to which it adheres laterally, and, finally, attaches just distal to the lateral aspect of the radial notch of the ulna (Figure 14C, no. 4). It is well visible that the articular capsule (Figure 14A,B, no. 5) unifies with the lateral collateral ligament. It is only after complete maceration of the articular capsule that the true nature of the merged annular and lateral collateral ligament is revealed. Distal to the lateral collateral ligament can the oblique chord (*chorda obliqua*) (Figure 14B and insert, no. 6) be seen. This structure extends from the cranial side of the ulna to the caudal side of the radius, describing a diagonal course. Distal to it can the antebrachial interosseous membrane (*membrana interossea antebrachii*) (Figure 14B insert, no. 7) be seen connecting the ulnar and radial diaphyses in proximodistal direction from the ulna to the radius and vice versa. A glimpse of the inserting tendon of the biceps brachii muscle is visible in the lateral view.

The medial view is particularly suited to study the insertions of the biceps brachii and the brachialis muscles. It was already mentioned that the medial collateral ligament of the elbow joint consists of a proximal part (Figure 15, no. 1a) and a distal part (Figure 15, no. 1b). The common trunk is initiated at the medial humeral epicondyle (Figure 15A–C, no. 2). The articular capsule (Figure 15A,B, no. 3) follows the contours of the humeral trochlea (Figure 15C, no. 4). Both parts of the medial collateral ligament separate at this level. Unlike the lateral collateral ligament of the elbow joint, the medial collateral ligament is not merged with the articular capsule but overlies it. The inserting tendon of the brachialis muscle (Figure 15A–C, no. 5) runs through the window between both parts of the medial collateral ligament to attach to the ulnar tuberosity (*tuber ulnae*). The biceps brachii muscle (Figure 15A–C, no. 6), on the other hand, attaches to radial tuberosity and the antebrachial interosseous membrane, immediately distal to the insertion of the brachialis muscle.

### 3.5. Hand

#### 3.5.1. General Conformation

The distal epiphyses of the radius and ulna form the distal radioulnar joint (*art. radioulnaris distalis*) (Figure 16B, no. 1), which is, however, not included in the joints of the hand (*articulationes *(*arts.*)* manus*). This term is the heading for all joints of the hand.

The joints that are formed between the antebrachial bones (radius and ulna) and the proximal row of carpal bones between the carpal bones themselves, between the proximal and distal rows of the carpal bones, and between the pisiform bone, on the one hand, and the ulna and triquetrum bones, on the other hand, form the carpal joint (*art. carpi*). The proximal carpal or antebrachiocarpal joint (*art. antebrachiocarpea*) (Figure 16B, no. 2) connects the distal epiphyses of the radius and ulna with the proximal or antebrachial row of carpal bones, i.e., form medial to lateral, the scaphoid bone (*os scaphoideum* or *os carpi radiale*), lunate bone (*os lunatum* or *os carpi intermedium*), and triquetrum bone (*os triquetrum* or *os carpi ulnare*). More specifically, the radiocarpal joint (*art. radiocarpea*) (Figure 16B, no. 3) is the connection between the carpal articular surface (*facies articularis carpea*) of the radius, on the one hand, and the scaphoid and lunate bones, on the other hand, while the carpal articular surface at the head of the ulna (*caput ulnae*) forms the ulnocarpal joint (*art. ulnocarpea*) (Figure 16B, no. 4) with the triquetrum bone. The intercarpal joints (*arts. intercarpeae*) (Figure 16B, no. 5) can be found in between the several carpal bones. The distal carpal joint (*art. mediocarpea*) (Figure 16B, no. 6), which is also a condyloid joint like the proximal carpal joint, mediates the connection between the proximal and distal rows of carpal bones. From medial to lateral, the distal or metacarpal row is composed of the trapezium bone (*os trapezium* or *os carpale primum*), trapezoid bone (*os trapezoideum *or* os carpale secundum*), capitate bone (*os capitatum* or *os carpale tertium*), and hamate bone (*os hamatum* or *os carpale quartum*). The central carpal bone articulates with the scaphoid and lunate of the proximal row, as well as with the trapezium, trapezoid, and capitate of the distal row. The pisiform bone (*os pisiforme* or *os carpi accessorium*) presents its own joint (*art. ossis pisiformis/carpi accessorii*) (Figure 16B, no. 7). It is in contact with the styloid process (*processus styloideus*) of the ulna proximally and the triquetrum bone distally.

The carpometacarpal joints (*arts. carpometacarpeae*) (Figure 16B, no. 8) form the connection between the distal row of carpal bones and the metacarpal bones. To be specific, the trapezoid bone articulates with the second metacarpal bone, the capitate bone with both the second and third metacarpal bones, and the hamate bone with both the fourth and fifth metacarpal bones. The carpometacarpal saddle joint of the thumb (*art. carpometacarpea pollicis*) (Figure 16B, no. 9) is formed between the trapezium bone and the first metacarpal bone.

The three intermetacarpal joints (*arts. intermetacarpeae*) (Figure 16B, no. 10) form the connection between the neighboring metacarpal bones II, III, IV, and V. The metacarpal bone for the thumb has no contact with the second metacarpal bone. Its only contact is with the trapezium bone.

More distally, the metacarpophalangeal joints (*arts. metacarpophalangeae*) (Figure 16B, no. 11) connect the heads of the metacarpal bones with the bases of the proximal phalanges. Consequently, they are five in number.

The interphalangeal joints form the congruence between the phalanges. Except for the thumb, both a proximal interphalangeal joint (*art. interphalangea proximalis manus*) (Figure 16B, no. 12) and a distal interphalangeal joint (*art. interphalangea distalis manus*) (Figure 16B, no. 13) can be found in each digit. Since the first digit or thumb (*pollex*) consists of only two phalanges, i.e., the proximal and the distal, while the other four digits are composed of three phalanges, it presents a single interphalangeal joint (*art. interphalangea pollicis*) (Figure 16B, no. 14).

#### 3.5.2. Myology

##### Dorsal Approach

After skinning the dorsal side of the carpus and hand, various tendons whose positions are secured by reinforcements of the deep fascia (*fascia dorsalis manus*) (Figure 17A, no. 1) can be observed. This so-called extensor retinaculum (*retinaculum extensorum*), previously known as the dorsal carpal ligament (*lig. carpi dorsale*) is double. It consists of the proximal extensor retinaculum (*retinaculum extensorum proximale*) (Figure 17A, no. 2) and the distal extensor retinaculum (*retinaculum extensorum distale*) (Figure 17A, no. 3). In the text below, the muscles with their tendons that travel at the dorsal side of the hand and traverse the carpus are described. To this purpose, the extensor retinaculum was transected (Figure 17B,C).

The abductor digiti primi/pollicis longus muscle (*m. abductor digiti primi/pollicis longus*) (Figure 17A–C, no. 4) starts at the dorsal surfaces of the antebrachial bones and the interosseous membrane. It runs deep, underneath the extensor digitorum communis muscle (Figure 17A–C, no. 5) and ends at the lateral side of the first metacarpal bone, just distal to the carpometacarpal joint of the thumb. A sesamoid bone (*os sesamoideum m. abductoris digiti primi/pollicis longi*) is embedded in the tendon where it passes the medial side of this joint (Figure 17A–C, no. 6).

Like the other extensors of the carpus and digits, the extensor digiti primi longus muscle (*m. extensor digiti primi/pollicis longus*) (Figure 17A–C, no. 7) starts at the lateral epicondyle of the humerus. It runs in between the radius and ulna to the proximal phalanx of the thumb. The extensor digitorum secundi et tertii muscle (*m. extensor digitorum secundi et tertii*) (Figure 17A–C, no. 8) starts with a single muscle belly at the lateral humeral epicondyle. It splits in two individual tendons that attach to the proximal phalanges of the second and third digits. These tendons are fully exposed after the distal retraction of the common tendon of the extensor digitorum communis muscle (Figure 17C, no. 5). This manipulation demonstrates that the common tendon branches in four tendons that insert into the distal phalanges of the second to fifth digits. The extensor digitorum quarti et quinti muscle (*m. extensor digitorum quarti et quinti*) (Figure 17A–C, no. 9) has an analogous origin and insertion as the extensor digitorum secundi et tertii muscle, initiating as a single muscle belly from the lateral humeral epicondyle, before splitting into two tendons, one for each of the proximal phalanges of the fourth and the fifth digits.

The tendons of the extensor carpi radialis brevis muscle (Figure 17A–C, no. 10), the extensor carpi radialis longus muscle (Figure 17A–C, no. 11), and the extensor carpi ulnaris muscle (Figure 17A–C, no. 12) remain after resection of the above-described muscle tendons. The short extensor radialis muscle inserts into the dorsal surface of the base of the second metacarpal bone, whereas the tendon of the long extensor radialis muscle has a similar attachment, albeit at the third metacarpal bone. The tendon of the extensor carpi ulnaris muscle inserts into the dorsal aspect of the base of the fifth metacarpal bone.

Since the here-described muscles pass over the various joints of the hand, they have to be resected to gain dorsal access to these underlying joints. Their articular capsules are visible in Figure 17D (no. 13).

##### Palmar Approach

Skinning the palmar side of the carpus and hand unveils the deep fascia of the palm of the hand (*fascia palmaris manus*) (Figure 18A, no. 1), which is reinforced at the level of the distal antebrachium and carpus. This flexor retinaculum (*retinaculum flexorum*) (Figure 18A, no. 2), previously known as the transverse carpal ligament (*lig. carpi transversum*), covers the flexor muscles of the carpus and digits at the palmar side of the wrist. It retains their tendons in position by forming the carpal tunnel (*canalis carpi*). The pisiform bone is an important anchor point of the flexor retinaculum. Transecting this flexor retinaculum in the sagittal plane visualizes the tendons running underneath it. The tendon of the palmaris longus muscle (Figure 18B, no. 3), which has its origin on the medial humeral epicondyle, is revealed first. This tendon terminates in the palmar aponeurosis (*aponeurosis palmaris*) (Figure 18A,B, no. 4). The distal aspect of this aponeurosis presents a transverse strengthening called the superficial transverse metacarpal ligament (*lig. metacarpale transversum superficiale*) (Figure 18A,B, no. 5). When the palmar aponeurosis together with the palmaris longus muscle is removed, the deeper layer of the flexor retinaculum is uncovered (Figure 18C, no. 2).

The deeper layer of the flexor retinaculum is the origin of numerous smaller muscles of the first and fifth digits. The intrinsic muscles of the thumb include, from medial to lateral in the superficial layer, the abductor digiti primi/pollicis brevis muscle (*m. abductor digiti primi/pollicis brevis*) (Figure 18A–C, no. 6) and the superficial head of the flexor digiti primi/pollicis brevis muscle (*m. flexor digiti primi/pollicis brevis caput superficiale*) (Figure 18A–C, no. 7). In the deeper layer lie, also from medial to lateral, the opponens digiti primi/pollicis muscle (*m. opponens digiti primi/pollicis*) (Figure 18A–C, no. 8), the deep head of the flexor digiti primi/pollicis brevis muscle (*m. flexor digiti primi/pollicis brevis caput profundum*) (Figure 18D, no. 9), and the adductor digiti primi/pollicis muscle (*m. adductor digiti primi/pollicis*) (Figure 18D, no. 10). The fifth digit presents, in the superficial layer from lateral to medial, the abductor digiti quinti/minimi muscle (*m. abductor digiti quinti/minimi*) (Figure 18A–C, no. 11) and the flexor digiti quinti/minimi muscle (*m. flexor digiti quinti/minimi*) (Figure 18A–C, no. 12). The opponens digiti quinti/minimi muscle (*m. opponens digiti quinti/minimi*) (Figure 18D, no. 13) lies in the deeper layer at the axial side of the fifth metacarpal bone.

Resecting the deeper layer of the flexor retinaculum and the superficial layer of the intrinsic muscles of the first and fifth digits uncovers the tendon of the flexor digitorum superficialis muscle (Figure 18D,E, no. 14). This muscle is initiated with a single belly at the medial epicondyle of the humerus and passes into a broad common tendon. At the level of the carpus, the common tendon splits in four tendons that run to the middle phalanges of the second to fifth digits. The three heads of the flexor digitorum profundus muscle (Figure 18D–F, no. 15) unite into a broad, flat tendon at the level of the carpus. This tendon runs deep to the common tendon of the superficial digital muscle. At the level of the metacarpal bones, the tendon divides into five tendons that attach to the distal phalanges of all five digits. The lumbrical muscles of the hand (*mm. lumbricales manus*) (Figure 18E, no. 16) can be visualized onto the palmar side of the tendon of the deep digital flexor muscle, at the transition of the common tendon to the individual tendons. They end on the proximal phalanges of the second to fifth digits and their metacarpophalangeal joint capsules.

The deeper layer is reached after retracting the tendons of the deep digital flexor in the distal direction. The palmar interosseous muscles (*mm. interossei manus palmares*) and the contrahentes muscles of the hand (*mm. contrahentes digitorum manus*) remain in the palm of the hand, deep against the second to fifth metacarpal bones (Figure 18E, no. 17). Removing these muscles provides access to the dorsal interosseous muscles (*mm. interossei manus dorsales*) (Figure 18F, no. 18). At the distal antebrachium, the pronator quadratus muscle (*m. pronator quadratus*) (Figure 18F, no. 19) also comes into sight.

Finally, the flexor carpi ulnaris muscle (Figure 18A–F, no. 20) and the flexor carpi radialis muscle (Figure 18A–E, no. 21) should be mentioned here. The former is initiated on the medial humeral epicondyle and distally attaches to the proximal aspect of the pisiform bone. The latter has the same origin but inserts into the base of the second metacarpal bone.

#### 3.5.3. Arthrology

After the removal of all the muscles and tendons at the dorsal side of the hand, the articular capsule of the carpal joint (Figure 19A, no. 1) is visible. Just proximal, the articular capsule of the distal radioulnar joint (Figure 19A, no. 2) can be observed. Maceration of the soft tissues reveals several ligaments embedded in the joint capsules. The dorsal radioulnar ligament (*lig. radioulnare dorsale*) (Figure 19C, no. 3) presents an almost transverse fiber direction on the dorsal side of the distal radioulnar joint capsule. It secures the ulnar head against the ulnar notch of the radius. The analogous ligament at the palmar side of the distal radioulnar joint capsule (Figure 20A, no. 1) is the palmar radioulnar ligament (*lig. radioulnare palmare*) (Figure 20A,B, no. 2).

Regarding the antebrachiocarpal joint, and more specifically the radiocarpal joint, the dorsal radiocarpal ligament (*lig. radiocarpeum dorsale*) (Figure 19C, no. 4) is by far the larger of the two ligaments at the dorsal side. It extends from the transverse ridge at the dorsal side of the distal epiphysis of the radius to the dorsal aspects of the lunate and triquetrum bones. The medial collateral ligament of the wrist joint (*lig. collaterale mediale/radiale carpi*) (Figure 19C, no. 5) is much smaller. It runs from the dorsal side of the styloid process of the radius to the dorsolateral side of the scaphoid bone. It is a lateral reinforcement of the radiocarpal joint capsule (Figure 19B, no. 6). At the palmar side, the palmar radiocarpal ligament (*lig. radiocarpeum palmare*) (Figure 20B, no. 3) runs from the palmar side of the styloid process of the radius to the palmar aspects of the scaphoid and capitate bones.

When considering the ulnocarpal joint, is it observed that the joint capsule (Figure 19B, no. 7) is laterally reinforced by the lateral collateral ligament of the wrist (*lig. collaterale laterale/ulnare carpi*) (Figure 19C, no. 8). It runs from the dorsolateral side of the styloid process of the ulna to the dorsolateral sides of the triquetrum and pisiform bones. The palmar ulnocarpal ligament (*lig. ulnocarpeum palmare*) (Figure 20A,B, no. 4) is part of the ulnocarpal complex. This complex connects the palmar side of the head of the ulna with the palmar sides of several carpal bones. The palmar ulnocarpal ligament crosses the palmar radiocarpal ligament to attach to the palmar sides of the lunate and trapezium bones. Other ligaments of this complex are smaller and innominate. It is worthwhile to mention the ligament between the styloid process of the ulna and the pisiform bone (*lig. ulnopisiforme*) (Figure 20B, no. 5).

Still at the palmar side of the carpus, a ligament runs from the medial side of the pisiform bone to the palmar side of the central carpal bone (*lig. pisocentrale*) (Figure 20A,B, no. 6), while another ligament attaches to the palmar side of the hamate bone. This is the pisohamate ligament (*lig. pisohamatum*) or accessorioquartal ligament (*lig. accessorioquartale*) (Figure 20A,B, no. 7). The pisometacarpal ligament (*lig. pisometacarpeum*) or accessoriometacarpal ligament (*lig. accessoriometacarpeum*) (Figure 20A,B, no. 8) runs laterodistal to the pisohamate ligament, connecting the pisiform bone with the base of the fifth metacarpal bone. At the medial side of the carpus remains the radiate carpal ligament (*lig. carpi radiatum*) (Figure 20B, no. 9) to be mentioned. It forms a vast palmar structure that connects all adjacent structures. It attaches to the palmar ulno- and radiocarpal ligaments, pisometacarpal ligament, multiple carpal bones, and the interosseus muscles.

In addition to the above-mentioned ligaments that are located at the level of the intercarpal joints, dorsal intercarpal ligaments (*ligg. intercarpea dorsalia*) (Figure 19C, no. 9) can be recognized at the dorsal side of the carpus. A vast number of ligaments start from the triquetrum bone and run medially to the lunate bone, the distal aspect of the scaphoid bone, and, finally, the trapezium bone. Together with the dorsal radiocarpal ligament, they form a V with the tip at the triquetrum bone. Additional smaller and individual ligaments can be observed more distal.

Numerous dorsal carpometacarpal ligaments (*ligg. carpometacarpea dorsalia*) (Figure 19B, no. 10) can be found at the dorsal sides of the carpometacarpal joints. They connect the distal row of carpal bones with the five metacarpal bones. At the palmar side, similar structures called palmar carpometacarpal ligaments (*ligg. carpometacarpea palmaria*) (Figure 20B, no. 10) can be identified.

The intermetacarpal joints are provided with both dorsal and palmar ligaments. The dorsal metacarpal ligaments (*ligg. metacarpea dorsalia*) (Figure 19B,C, no. 11) run from the base of each metacarpal bone to the base of the adjacent metacarpal bone and also connect with the carpometacarpal ligaments. At the palmar side, the palmar metacarpal ligaments (*ligg. metacarpea palmaria*) (Figure 20B, no. 11) show a similar trajectory.

In the male animal that was dissected, an additional carpal bone not found in the other two specimens was encountered at the level of the antebrachiocarpal joint. It was located between the styloid process of the ulna and the triquetrum bone (encircled structure in Figure 21A). Proximally, it was attached to the dorsal radioulnar ligament and distally to the lateral collateral ligament. When it was first detached from this ligament (Figure 21B) and, subsequently, from the dorsal radioulnar ligament, its ovoid shape could be noticed (Figure 21C).

The metacarpophalangeal joint capsules (Figure 22A, no. 1) are reinforced at their axial and abaxial sides by collateral ligaments (*ligg. metacarpophalangea collateralia)* (Figure 22A, no. 2, and Figure 23A–C, no. 1). These ligaments have palmar branches (Figure 23A–C, no. 2) that attach to the pair of ovoid sesamoid bones (*ossa sesamoidea palmaria proximalia*) (Figure 23A–C, no. 3) that sits at the palmar side of each metacarpal trochlea. The palmar metacarpophalangeal ligaments (*ligg. metacarpophalangea palmaria*) (Figure 22B, no. 3, and Figure 23A–C, no. 4) provide the tight connection between the axial and abaxial sesamoid bone of each pair. The deep transverse metacarpal ligaments (*ligg. metacarpalia transversa profunda*) (Figure 22B, no. 4) connect the axial sesamoid bones of the third and fourth digit, and the respective abaxial and axial sesamoid bones of the fourth and fifth digits, on the one hand, and the third and second digits, on the other hand. These three ligaments, thus, bridge the digits, with the exception of the thumb.

The ligaments of the proximal and distal interphalangeal joints are highly comparable. Collateral interphalangeal ligaments (*ligg. interphalangea collateralia*) (Figure 24A,B, no. 1, and Figure 25A–D, no. 1) are present at the lateral and medial sides of each interphalangeal joint. In addition, palmar interphalangeal ligaments (*ligg. interphalangea palmaria*) (Figure 24A,B, no. 2) are present in the proximal interphalangeal joint. They are palmar branches of the collateral ligaments that attach to the distal palmar sesamoid bone (*os sesamoideum palmare distale*) (Figure 24A,B, no. 3).

## 4. Discussion

This manuscript presents the anatomy of the synovial joints of the thoracic limb of the hamadryas baboon. The various joints were dissected in three animals, taking color photographs during the subsequent steps. The observations were textually described aiming for the use of this work as an anatomical atlas or dissection guide. The rationale behind this approach can be found in the fact that veterinarians responsible for the medical care of baboons are confronted with the lack of comprehensive species-specific anatomical data. This should not be interpreted as if there are no data at all. Numerous studies have been published examining particular aspects of the joints of the baboon. These studies are often very in-depth and, therefore, fail to provide the general conformation of the joint under study. In addition, since baboons are wildlife species, their anatomy is not taught in the standard curriculum of veterinary medicine. The only comprehensive work on baboon anatomy is the atlas by Swindler and Wood [6]. It shows many illustrations of various anatomical systems. Regrettably, the quantity of supporting text is restricted. In addition, only black-and-white line drawings that fail to recapitulate the anatomical details are included. As a result, doubtful identifications might appear during dissections or surgical interventions. However, all of this is irrelevant in this specific context, since the arthrology is not included in the Swindler and Wood atlas [7]. The reason for this can only be guessed at. Own experience has learned that the study of this anatomical system is challenging, though. Veterinarians working with primates, thus, fall back on human anatomy atlases, a choice that is based on the great similarity between humans and other primates that has its origin in their close genetic relationships [12]. Below, a number of striking differences between baboon and human anatomy are briefly reviewed. These potentially find their origins in the way the thoracic limb is used, i.e., focus on locomotion and stabilization in genus *Papio* vs. focus on practical usage and fine motor dexterity in man. The aim of the enumeration is to underscore that the consultation of human anatomical works can be misinforming when other primate species, in case of the baboon, are under study. It should be clear that baboon-specific anatomical data can contribute to the welfare of baboons requiring medical attention.

The dissimilarities observed in the shoulder region are considered first, starting with the myology. The dorsoepitrochlearis muscle of the baboon arises from the ventral margin of the latissimus dorsi muscle and merges with the long head of the triceps brachii muscle. During evolution, it has fused with the latissimus dorsi muscle. It is, consequently, absent in most humans, although the muscle can be encountered as an anatomical variation [18].

The baboon coracobrachialis muscle is composed of the deep and the middle coracobrachialis muscles. Among the domestic mammals, a similar division is also present in the horse, ruminants, and the rabbit [16]. In these species, a proximal, deep part and a distal, superficial part are present. In contrast, it is a single muscle in the larger majority of humans [19,20].

The pectoralis musculature slightly differs in morphology and nomenclature between the baboon and man. The baboon pectoralis superficialis and profundus muscles are largely the analogues of the pectoralis major and minor muscles in humans, respectively. However, the sternocapsular part of the superficial pectoral muscle as seen in the baboon is more developed in humans. Only a very limited attachment to the clavicle is present in the baboon, whereas the medial third of the clavicle is the attachment site of the clavicular part (*pars clavicularis*) in man [20].

Moving on to the arthrology, it was observed that the transverse humeral ligament secures the tendon of the biceps brachii muscle in the intertubercular groove. In human anatomy, the true nature of this ligament is under debate. It could be regarded as a mere continuation of the tendon fibers of the subscapularis and pectoralis superficialis muscles [21]. Although it is included in the T.A. [14], human anatomy atlases including the highly esteemed Sobotta [20,22] do not mention the name. In contrast, it is a well-developed ligament in domestic mammals [23]. However, the term *ligamentum transversum humeri* has been replaced by the term *retinaculum transversum* [13].

As the clavicle is present in both baboons and humans, it is expected that similar ligaments securing the position of this bone can be encountered. In humans, the coracoclavicular ligament consists of the trapezoid ligament (*lig. trapezoideum*) and the conoid ligament (*lig. conoideum*) [20]. The former attaches to the basis of the coracoid process whereas the latter has the transition of the scapular neck to the basis of the coronoid process as its attachment site. Such bipartite ligament was not observed in the dissected specimens. Only a single ligament resembling the human trapezoid ligament could be identified. A costoclavicular ligament was not observed in the dissected baboons. In contrast, such ligament supports the sternoclavicular joint in apes and humans in which it limits horizontal and vertical clavicle movement. Consequently, its absence allows for a greater mobility of the thoracic limb in the baboon [24]. Since no or only a reduced clavicle is present in domestic mammals, neither of these ligaments can be observed in these quadrupedal species. In quadrupedal primates possessing a clavicle such as baboons, this bone allows for the upper limb to realize movements outside the sagittal plane, without hindering quadrupedal locomotion [25]. The latter probably is the reason why the baboon has an intermediate position, in terms of the presence of clavicular ligaments, in between man and domestic mammals.

Furthermore, the coracoacromial ligament (*lig. coracoacromiale*) spans between the tip of the coracoid process and the tip of the acromion in man. It was not found in any of the dissected specimens. In contrast, the coracohumeral ligament extending between the coracoid process and the greater tubercle is present in both the baboon and man.

The elbow joint is subsequently considered, reflecting first on the myology. A few minor dissimilarities with man are communicated first. In contrast to humans, the baboon flexor carpi radialis muscle only consists of the humeral head and no ulnar head could be observed. This was also described earlier [6]. Also, the anconeus muscle in man is very short. It spans from the lateral humeral condyle to the olecranon. The anconeus muscle in the baboon is well developed. It arises from the caudal side of the distal third of the humerus and attaches to the anconeal process. The muscle in the baboon that closely resembles the human anconeus muscle is the epitrochleoanconeus muscle. However, this muscle is located at the medial side of the elbow joint. The prevalence in a central European population is only 4% [26].

The insertions of the biceps brachii and brachialis muscles deserve sufficient attention. According to the description of genus *Papio* by Swindler and Wood [6], the biceps brachii muscle inserts into the radial tuberosity. This is also the case in humans [20,27]. However, the distal tendons of the biceps brachii muscles in the dissected baboons inserted into the radial tuberosity and the interosseous membrane. This membrane seems a secondary attachment site for the biceps brachii muscle in the baboon. In humans, the distal tendon of the biceps brachii muscle presents an aponeurotic expansion toward the antebrachial fascia (*aponeurosis bicipitis brachii*) [20]. A comparable condition appears in domestic mammals, in which this so-called *lacertus fibrosus* that is particularly well-developed in ungulates unites with the extensor carpi radialis muscle and not with the antebrachial fascia [19]. This structure is, thus, located at the medial side of the human arm. In contrast, it lies at the lateral side of the thoracic limb is domestic mammals. Surprisingly, such structure was not identified in the baboon during the present study. Further examinations specifically targeting this matter could potentially provide more insight. It was observed that the inserting tendon of the baboon’s brachialis muscle attaches to the ulnar tuberosity, which is located distal to the medially oriented coronoid process of the ulna. This process serves as the distal site of attachment for the anterior part of the ulnar (medial) collateral ligament in humans. The posterior part not only attaches posterior to it but also somewhat more proximal. Both parts are, however, tied to each other [20]. In the dissected baboons, the medial collateral ligament of the elbow joint consisted of a proximal part and a distal part, attaching proximal and distal to the ulnar tuberosity, respectively. In between both, the tendon of the brachialis muscle inserted into the ulnar tuberosity.

In a study assessing terminal elbow extension in the olive baboon, the role of the anterior oblique ulnar collateral ligament bundle was investigated [28]. First, it is advised to use veterinary nomenclature, thus using the term cranial oblique medial collateral ligament. The authors also mention a middle part of the ulnar collateral ligament. This might suggest that a caudal part is additionally present. In humans and hamadryas baboons, an anterior/cranial and a posterior/caudal part are present [20]. No caudal part is visible on the photographs shown by the authors. As a consequence, that middle part probably represents the caudal part. Further complicating this matter, it should be noticed that the T.A. only mentions the *ligamentum collaterale ulnare cubiti* in humans without detailing the existence of an anterior and a posterior part [14]. In contrast, the Sobotta does [20]. Since the cranial part of the ulnar collateral ligament attaches more distally on the ulna, it shows an oblique course. As a result, it could be understood that the anterior ulnar collateral ligament and the oblique ulnar collateral ligament are synonyms. Thus, no posterior oblique ulnar collateral ligament exists. This point of discussion makes it clear that the use of standardized anatomical nomenclature is pivotal for the correct interpretation of studies.

Finally, some thoughts on the articulations of the hand are communicated. In both the baboon and man, the carpal articular surface of the radius rests on the scaphoid and lunate bones. A proper congruency is present between these bones. However, it was previously stated that baboons are the only taxon in which the dorsal lip of the scaphoid does not appear to fully engage with the bony lip of the dorsal radius, which is in contrast to what is observed in palmigrade-capable monkeys to which baboons belong [29]. As a result of this incongruency, a semilunar meniscus sitting in the dorsolateral aspect of the radiocarpal joint is described in baboons [30]. Unfortunately, no unambiguous images of such meniscus are available in literature. To be precise, only a black-and-white photograph of a scaphocentrale-triquetral ligament originating from the presumed ill-visible semilunar meniscus and attaching to the scaphoid, central and triquetrum bones can be found in literature. This ligament seems to correspond to the more proximal ligament of the dorsal ligamentous complex that is called the *ligg. intercarpea dorsalia* in the present manuscript. A semilunar meniscus could not be isolated from that ligament in the present study. It is suggested that the meniscus is a mere reinforcement of the origin of the ligament on both the bony lip of the dorsal radius and the scaphoid bone. In contrast, the head of the ulna is not fitting the triquetrum bone in man. As a consequence, an articular disc provides the congruency [20]. Such a disc was not encountered in the dissected specimens. The reason might be found in the elongated ulnar styloid process in the baboon, in comparison with the poorly developed human counterpart. In the baboon, the ulnar styloid process seems to connect well with the triquetrum bone, excluding the need for an articular disc. However, the male hamadryas baboon that was dissected in the present study had an additional carpal bone that was not found in the other two specimens. Since it was located between the styloid process of the ulna and the triquetrum bone, it resembled an articular disc at first sight. Nevertheless, it was a bony structure and was attached to the dorsal radioulnar ligament proximally and to the lateral collateral ligament distally. In addition, it sat more in front of the joint between the ulnar styloid process and the triquetrum bone. It is, therefore, plausible that the structure was an ossification of the ligaments in which it was embedded. Future studies on more baboon hands could perhaps render more insight into the potential presence of an articular disc in the articulation between the ulnar styloid process and the triquetrum and/or pisiform bone. No other variations were found between the three dissected specimens. The low number of specimens dissected could be regarded as a limitation of our study. More specimens should be dissected in future studies when potential anatomical variations are to be discovered.

The robust ligaments that are characteristic for terrestrial primates disperse the ground forces over the entire hand [31]. In this regard, numerous studies have been published investigating the adaptation of the primate hand to the mode of locomotion. It should therefore not be surprising that manifold dissimilarities in the conformation of the baboon and human hand are present [31]. Reviewing the literature on this topic stretches beyond the scope of the present study. As a veterinarian, gaining insight into the specific build of the human hand is, nonetheless, superfluous and distracting when being confronted with arm and hand injuries in the baboon. The aim of the present work was to provide the veterinarian responsible for the medical care of baboons with anatomical data on the arthrology of this species’ thoracic limb should medical interventions be imposed.

## 5. Conclusions

The present manuscript provides anatomical data on the synovial joints of the thoracic limb of the hamadryas baboon. Photographs of the various joints, taken from various points of view, are shown to illustrate the textual descriptions. These start with an overview of the musculature surrounding the joint. Nonetheless, describing the entire muscular system of the mantled baboon in extenso was not the goal of the present manuscript. The focus was on the insertions and origins of muscles that are located in the proximity of the joints. A future study could revisit the musculature of the mantled baboon. Together with the previous manuscript on the osteology and the present and future manuscripts on the arthrology, a full atlas of the baboon’s locomotor system could emerge. During the dissections, the muscles were removed from the superficial to the deeper layers, ultimately reaching the joint. The structures associated with the joints, such as the ligaments and capsules, are subsequently discussed. This approach allows for the manuscript to be used as a dissection guide. This could be valuable during the tending of wounds or surgical interventions executed by the veterinarian responsible for the medical care of baboons. This consequently enhances the welfare of the animals.

## Figures and Tables

**Figure 1 animals-15-02894-f001:**
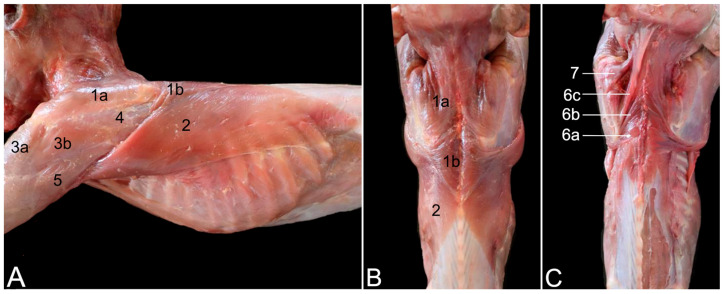
Lateral (**A**) and dorsal (**B**,**C**) views of the junctional musculature between the left thoracic limb and the trunk of the mantled baboon: (**A**) superficial layer; (**B**) superficial layer; (**C**) deeper layer after resection of the trapezius and latissimus dorsi muscles. 1a: m. trapezius pars cervicalis; 1b: m. trapezius pars thoracica; 2: m. latissimus dorsi; 3a: m. deltoideus pars acromialis; 3b: m. deltoideus pars spinalis scapularis; 4: m. infraspinatus covered by the aponeurosis of the deltoid muscle; 5: m. triceps brachii caput longum; 6a: m. rhomboideus thoracis; 6b: m. rhomboideus cervicis; 6c: m. rhomboideus capitis; 7: m. levator scapulae.

**Figure 2 animals-15-02894-f002:**
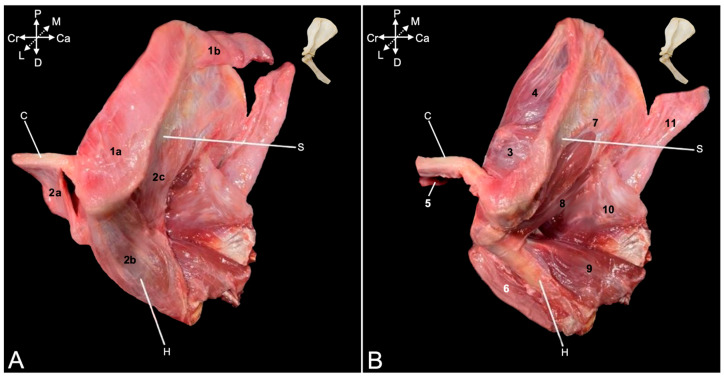
Lateral view of the musculature of the left shoulder and acromial region of the mantled baboon: (**A**) superficial layer; (**B**) deeper layer after resection of the trapezius and deltoid muscles. 1a: m. trapezius pars cervicalis; 1b: m. trapezius pars thoracica; 2a: m. deltoideus pars clavicularis; 2b: m. deltoideus pars acromialis; 2c: m. deltoideus pars spinalis scapularis; 3: m. omotransversarius; 4: m. supraspinatus; 5: m. subclavius; 6: m. biceps brachii; 7: m. infraspinatus; 8: m. teres minor; 9: m. triceps brachii caput laterale; 10: m. triceps caput longum; 11: m. latissimus dorsi; C: clavicula (lateral half); S: scapula; H: humerus (proximal half).

**Figure 3 animals-15-02894-f003:**
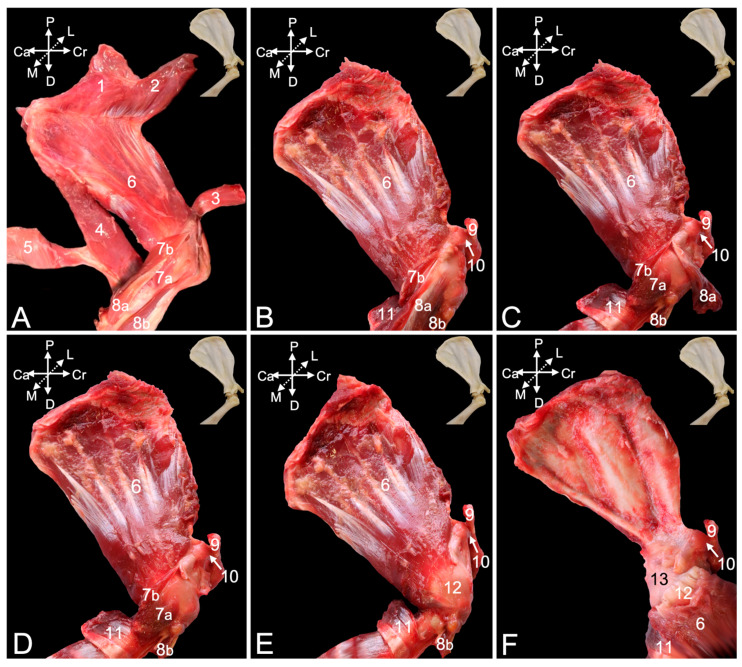
Medial view of the musculature of the left shoulder and acromial region of the mantled baboon: (**A**) superficial layer; (**B**) deeper layer after resection of the thoracic and cervical parts of the serratus ventralis muscle, subclavius muscle, and teres major and latissimus dorsi muscles; (**C**) deeper layer after transversely cutting and retracting the short head of the biceps brachii muscle; (**D**) deeper layer after resecting the short head of the biceps brachii muscle; (**E**) deeper layer after removing the deep and middle coracobrachialis muscles; (**F**) deepest layer after detaching the subscapularis muscle from its origin and distally retracting it. 1: m. serratus ventralis thoracis; 2: m. serratus ventralis cervicis; 3: m. subclavius; 4: m. teres major; 5: m. latissimus dorsi; 6: m. subscapularis; 7a: m. coracobrachialis profundus; 7b: m. coracobrachialis medius; 8a: m. biceps brachii caput breve; 8b: m. biceps brachii caput longum; 9: clavicula; 10: ligamentum coracoclaviculare; 11: stump of teres major and latissimus dorsi muscles; 12: tendon of subscapularis muscle; 13: medial side of the capsule of the shoulder joint.

**Figure 4 animals-15-02894-f004:**
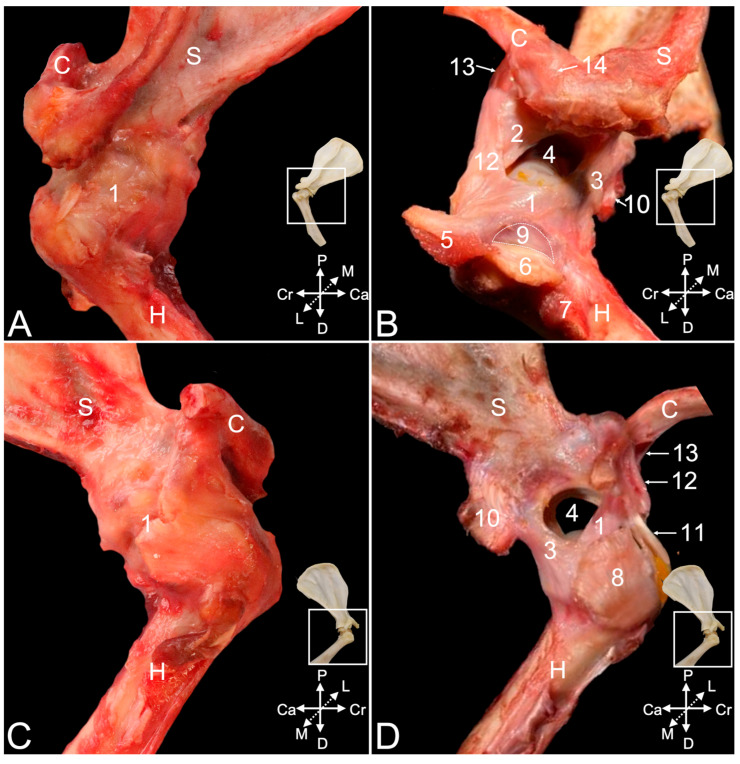
Arthrological structures of the left shoulder and acromial region of the mantled baboon: (**A**) lateral view before maceration; (**B**) lateral view after partial maceration; (**C**) medial view before maceration; (**D**) medial view after partial maceration. 1: capsula articularis; 2: lig. glenohumerale craniale; 3: lig. glenohumerale caudale; 4: cavitas articularis; 5: inserting tendon of m. supraspinatus; 6: inserting tendon of m. infraspinatus; 7: inserting tendon of m. teres minor; 8: inserting tendon of m. subscapularis; 9: bursa m. infraspinati (outlined); 10: originating tendon of the m. triceps brachii caput longum; 11: originating tendon of the m. biceps brachii caput longum; 12: ligamentum coracohumerale; 13: lig. coracoclaviculare; 14: lig. acromioclaviculare; C: clavicula (lateral half); S: scapula (distal half); H: humerus (proximal half).

**Figure 5 animals-15-02894-f005:**
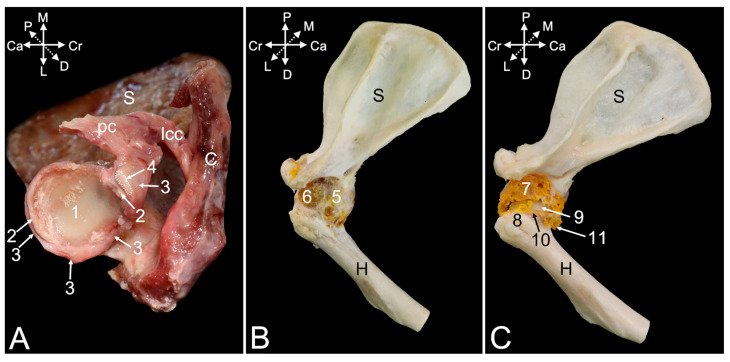
Arthrological structures of the left shoulder and acromial region of the mantled baboon: (**A**) ventral view of the glenoid cavity of the left shoulder blade before maceration; (**B**) lateral view of the left shoulder joint after maceration of the soft tissues and bleaching with H_2_O_2_, with polymethyl methacrylate cast of the joint cavity; (**C**) lateral view of the left shoulder joint after partial maceration of the polymethyl methacrylate cast. 1: cavitas glenoidalis; 2: labrum glenoidale; 3: capsula articularis; 4: originating tendon m. biceps brachii caput longum (outlined); 5: partially macerated capsula articularis with lig. glenohumerale caudale; 6: polymethyl methacrylate injectate in the articular cavity; 7: partially dissolved polymethyl methacrylate injectate; 8: tuberculum majus; 9: caput humeri; 10: collum humeri; 11: recessus axillaris; S: scapula; C: clavicula (lateral half); H: humerus (proximal half); lcc: lig. coracoclaviculare; pc: processus coracoideus.

**Figure 6 animals-15-02894-f006:**
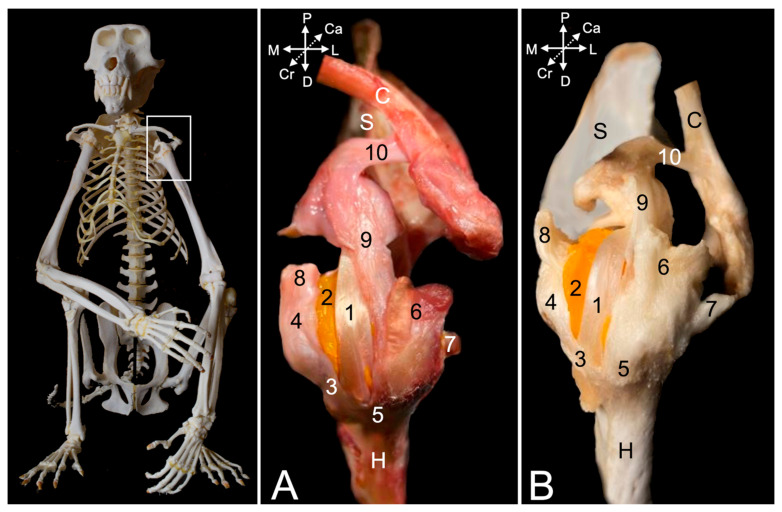
Cranial view of the arthrological structures of the left shoulder joint of the mantled baboon: (**A**) after removal of the muscles; (**B**) after maceration of the soft-tissue remnants and bleaching with H_2_O_2_. 1: tendon of the m. biceps brachii caput longum; 2: polymethyl methacrylate injectate in vagina tendinis intertubercularis; 3: lig. transversum humeri; 4: tuberculum minus; 5: tuberculum majus; 6: insertion of the m. supraspinatus; 7: insertion of the m. infraspinatus; 8: insertion of the m. subscapularis; 9: lig. coracohumeral; 10: lig. coracoclaviculare; S: scapula; C: clavicula (lateral half); H: humerus (proximal third).

**Figure 7 animals-15-02894-f007:**
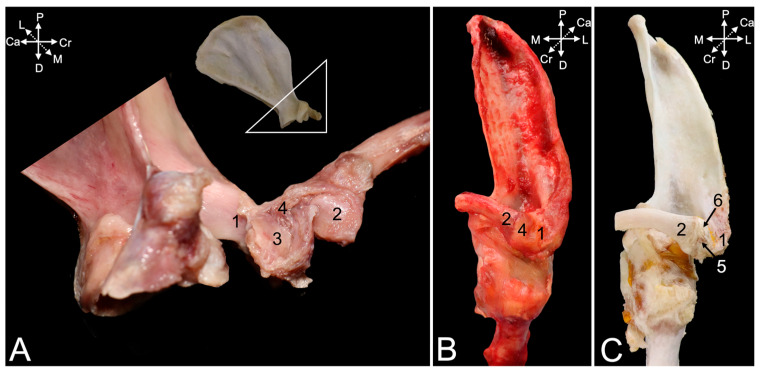
Arthrological structures of the left acromioclavicular joint of the hamadryas baboon: (**A**) medial view after partial maceration of the soft tissues; (**B**) cranial view before maceration; (**C**) cranial view after maceration of the soft-tissue remnants and bleaching with H_2_O_2_. 1: acromion; 2: extremitas acromialis; 3: discus articularis acromioclavicularis; 4: capsula articularis; 5: polymethyl methacrylate cast of the articular cavity; 6: ligamentum acromioclaviculare.

**Figure 8 animals-15-02894-f008:**
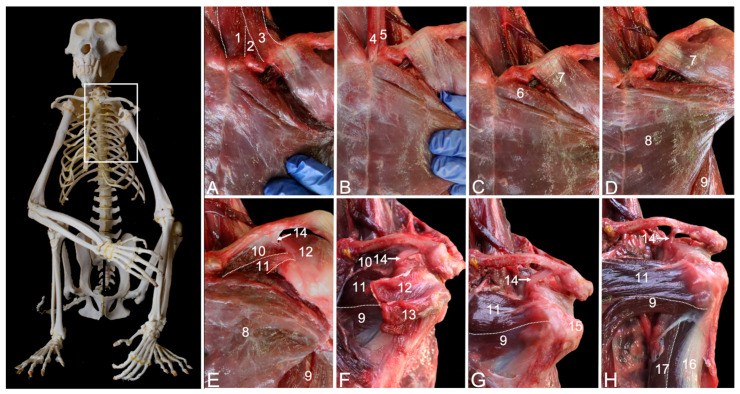
Ventral view of the musculature of the left sternal region of the mantled baboon: (**A**) superficial layer; (**B**) deeper layer after resection of the sternocleidomastoideus muscle; (**C**) deeper layer after resection of the sternohyoideus and sternothyroideus muscles; (**D**) deeper layer after resection of the sternoclavicular part of the superficial pectoral muscle; (**E**) deeper layer after resection of the clavicular part of the deltoid muscle; (**F**) deeper layer after resection of the sternal part of the superficial pectoral muscle, and retraction of the supraspinatus and infraspinatus muscles toward their insertions; (**G**) deepest layer after resection of the subclavius muscle, and the supraspinatus and infraspinatus muscles; (**H**) deepest layer with supination of the thoracic limbs demonstrating the biceps brachii muscle. 1: m. sternomastoideus; 2: m. cleidomastoideus; 3: m. cleidooccipitalis; 4: m. sternohyoideus; 5: m. sternothyroideus; 6: m. pectoralis superficialis pars sternocapsularis; 7: m. deltoideus pars clavicularis; 8: m. pectoralis superficialis pars sternalis; 9: m. pectoralis superficialis pars abdominalis; 10: m. subclavius; 11: m. pectoralis profundus; 12: m. supraspinatus; 13: m. infraspinatus; 14: lig. coracoclaviculare; 15: tuberculum majus; 16: m. biceps brachii caput longum; 17: m. biceps brachii caput breve.

**Figure 9 animals-15-02894-f009:**
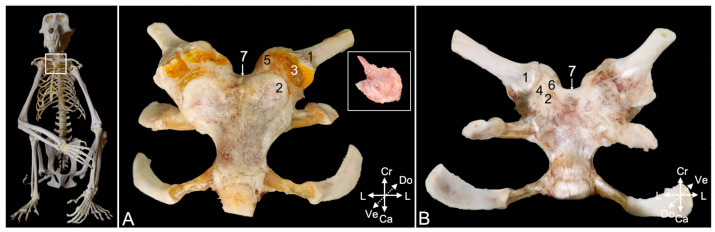
Cranial segment of the sternum of the hamadryas baboon, including the manubrium sterni, left and right clavicles, and left and right first two ribs (cut transversely). The sternoclavicular joints have been injected with polymethyl methacrylate. (**A**) Ventral view—the insert shows the discus articulationis sternoclavicularis. (**B**) Dorsal view—1: extremitas sternalis; 2: incisura clavicularis; 3: cavitas articularis; 4: capsula articularis; 5: lig. sternoclaviculare ventrale; 6: lig. sternoclaviculare dorsale; 7: lig. interclaviculare.

**Figure 10 animals-15-02894-f010:**
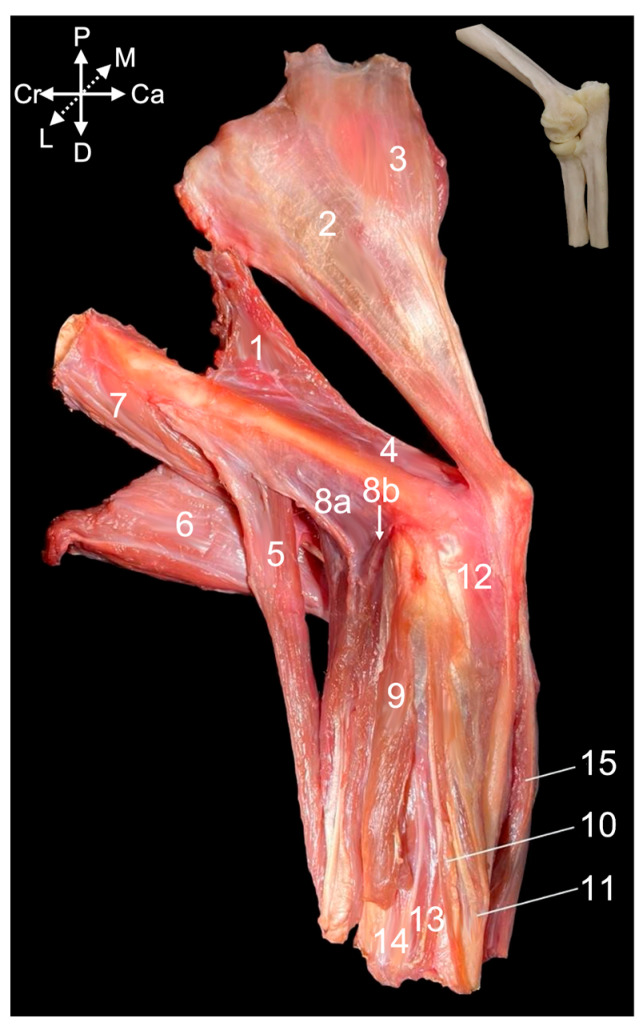
Lateral view of the musculature of the left elbow region of the hamadryas baboon: 1: m. triceps brachii caput laterale; 2: m. triceps brachii caput longum; 3: m. dorsoepitrochlearis; 4: m. anconeus; 5: m. brachioradialis; 6: m. biceps brachii; 7: m. brachialis; 8a: m. extensor carpi radialis longus; 8b: m. extensor carpi radialis brevis; 9: m. extensor digitorum communis; 10: m. extensor digiti quarti; 11: m. extensor digiti quinti/minimi; 12: m. extensor carpi ulnaris; 13: m. extensor digiti primi/pollicis longus; 14: m. abductor digiti primi/pollicis longus; 15: m. flexor carpi ulnaris.

**Figure 11 animals-15-02894-f011:**
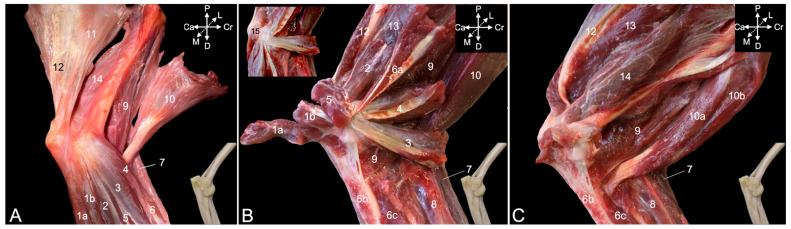
Medial view of the musculature of the left elbow region of the hamadryas baboon: (**A**) superficial layer. (**B**) deeper layer with transection and proximal retraction of the stumps of the pronator teres muscle and the flexor muscles of the carpus and digits and resection of the tensor fasciae antebrachii muscle. The insert shows a detail of the epithrochleoanconeus muscle. (**C**) Deepest layer after removal of the muscle stumps. 1a: m. flexor carpi ulnaris caput ulnare; 1b: m. flexor carpi ulnaris caput humerale; 2: m. palmaris longus; 3: m. flexor carpi radialis; 4: m. pronator teres; 5: m. flexor digitorum superficialis; 6a: m. flexor digitorum profundus caput humerale; 6b: m. flexor digitorum profundus caput ulnare (resected); 6c: m. flexor digitorum profundus caput radiale; 7: m. brachioradialis; 8: m. extensor carpi radialis longus; 9: m. brachialis; 10: m. biceps brachii; 10a: m. biceps brachii caput breve; 10b: m. biceps brachii caput longum; 11: m. tensor fasciae antebrachii; 12: m. triceps brachii caput longum; 13: m. triceps brachii caput mediale; 14: m. anconeus; 15: m. epithrochleoanconeus.

**Figure 12 animals-15-02894-f012:**
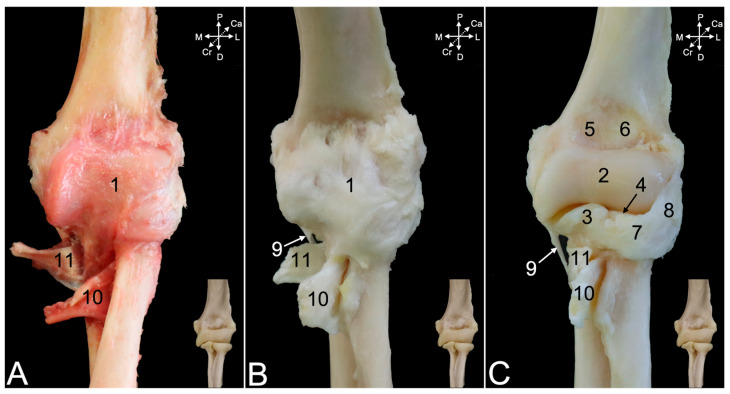
Cranial view of the left elbow joint of the hamadryas baboon: (**A**) after removal of the muscle stumps, with the exception of the inserting tendons of the biceps brachii and brachialis muscles; (**B**) after partial maceration of the soft-tissue remnants and bleaching with H_2_O_2_, leaving the articular capsule, joint ligaments, and insertions of the biceps brachii and brachialis muscles intact; (**C**) after further maceration of the articular capsule leaving the joint ligaments and insertions of the biceps brachii and brachialis muscles intact. 1: capsula articularis; 2: trochlea humeri; 3: processus coronoideus; 4: caput radii; 5: fossa coronoidea; 6: fossa radialis; 7: lig. anulare radii; 8: lig. colaterale cubiti laterale; 9: lig. collaterale cubiti mediale; 10: insertion of m. biceps brachii; 11: insertion of m. brachialis.

**Figure 13 animals-15-02894-f013:**
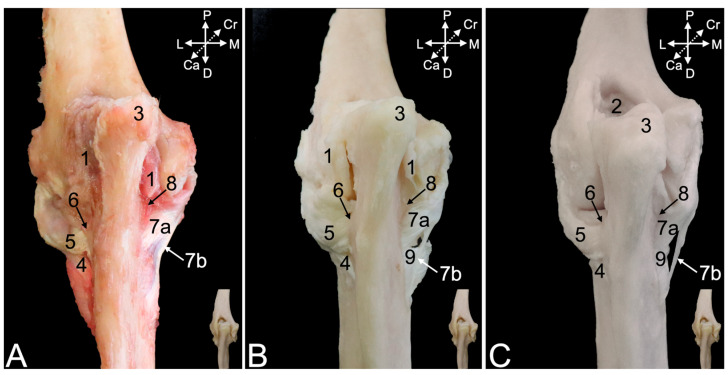
Caudal view of the left elbow joint of the hamadryas baboon: (**A**) after removal of the muscle stumps, with the exception of the inserting tendon of the brachialis muscle; (**B**) after partial maceration of the soft-tissue remnants and bleaching with H_2_O_2_, leaving the articular capsule, joint ligaments, and insertion of the brachialis muscle in place; (**C**) after maceration of the articular capsule leaving the joint ligaments and the insertion of the brachialis muscle intact. 1: capsula articularis; 2: fossa olecrani; 3: tuber olecrani; 4: lig. collaterale cubiti laterale; 5: lig. anulare radii; 6: incisura radialis; 7a: lig. collaterale cubiti mediale pars proximalis; 7b: lig. collaterale cubiti mediale pars distalis; 8: processus coronoideus; 9: insertion of m. brachialis.

**Figure 14 animals-15-02894-f014:**
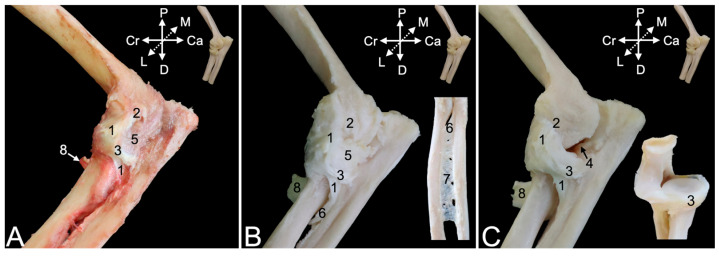
Lateral view of the left elbow joint of the hamadryas baboon. (**A**) After removal of the muscle stumps, with the exception of the inserting tendon of the biceps brachii muscle. (**B**) After partial maceration of the soft-tissue remnants and bleaching with H_2_O_2_, leaving the articular capsule, joint ligaments, and insertion of the biceps brachii muscle in place. The insert shows the chorda obliqua and the membrana interossei antebrachii. (**C**) After maceration of the articular capsule leaving the joint ligaments and the insertion of the biceps brachii muscle intact. The insert shows the lig. anulare radii. 1: lig. collaterale cubiti laterale; 2: epicondylus lateralis; 3: lig. anulare radii; 4: incisura radialis; 5: capsula articularis; 6: chorda obliqua; 7: membrana interossea antebrachii; 8: insertion of m. biceps brachii.

**Figure 15 animals-15-02894-f015:**
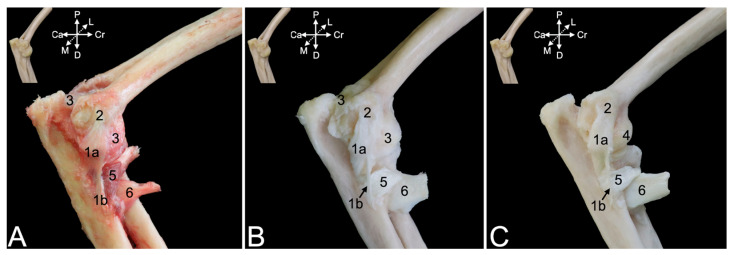
Medial view of the left elbow joint of the hamadryas baboon: (**A**) after removal of the muscle stumps, with the exception of the inserting tendons of the biceps brachii and the brachialis muscles; (**B**) after partial maceration of the soft-tissue remnants and bleaching with H_2_O_2_, leaving the articular capsule, joint ligaments, and insertions of the m. biceps brachii and the m. brachialis in place; (**C**) after maceration of the articular capsule leaving the joint ligaments and the insertions of the m. biceps brachii and the m. brachialis intact. 1a: lig. collaterale cubiti mediale pars proximalis; 1a: lig. collaterale cubiti mediale pars distalis; 2: epicondylus medialis; 3: capsula articularis; 4: trochlea humeri; 5: insertion of m. brachialis; 6: insertion of m. biceps brachii.

**Figure 16 animals-15-02894-f016:**
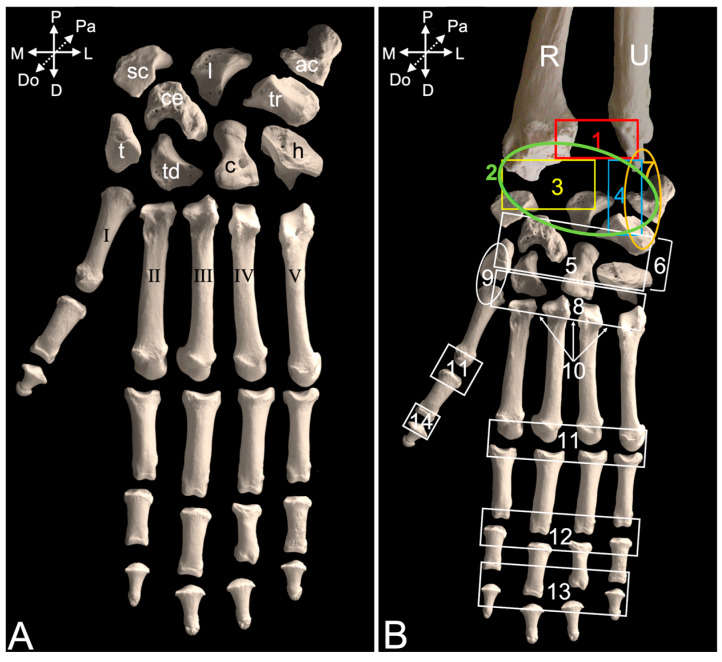
Dorsal view of the left hand skeleton of the hamadryas baboon: (**A**) the carpal and metacarpal bones are labeled; (**B**) the various articulations of the hand are indicated. sc: os scaphoideum/os carpi radiale; l: os lunatum/os carpi intermedium; tr: os triquetrum/os carpi ulnare; ac: os pisiforme/os carpi accessorium; ce: os carpi centrale; t: os trapezium/os carpale primum; td: os trapezoideum/os carpale secundum; c: os capitatum/os carpale tertium; h: os hamatum/os carpale quartum; 1 (red box): art. radioulnaris distalis; 2 (green circle): art. antebrachiocarpea; 3 (yellow box): art. radiocarpea; 4 (blue box): art. ulnocarpea; 5: arts. intercarpeae; 6: art. mediocarpea; 7 (orange circle): art. ossis pisiformis/carpi accessorii; 8: arts. carpometacarpeae; 9: art. carpometacarpea pollicis; 10: arts. intermetacarpeae; 11: arts. metacarpophalangeae; 12: art. interphalangea proximalis manus; 13: art. interphalangea distalis manus; 14: art. interphalangea pollicis; R: radius; U: ulna; I: os metacarpale primum; II: os metacarpale secundum; III: os metacarpale tertium; IV: os metacarpale quartum; V: os metacarpale quintum.

**Figure 17 animals-15-02894-f017:**
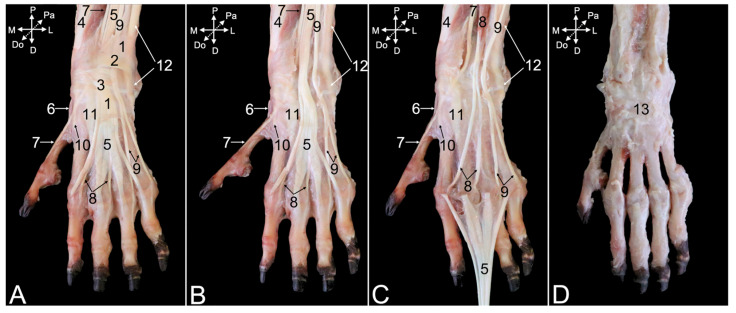
Dorsal view of the musculature of the left hand of the hamadryas baboon: (**A**) superficial layer with intact retinaculum extensorum; (**B**) superficial layer after transecting the retinaculum extensorum; (**C**) deeper layer after distal retraction of the m. extensor digitorum communis; (**D**) deepest layer after removal of all muscle tendons. 1: fascia dorsalis manus; 2: retinaculum extensorum proximale; 3: retinaculum extensorum distale; 4: m. abductor digiti primi/pollicis longus; 5: m. extensor digitorum communis; 6: inserting tendon of the m. abductor digiti primi/pollicis longus with embedded sesamoid bone; 7: m. extensor digiti primi/pollicis longus; 8: m. extensor digitorum secundi et tertii; 9: m. extensor digitorum quarti et quinti; 10: m. extensor carpi radialis brevis; 11: m. extensor carpi radialis longus; 12: m. extensor carpi ulnaris; 13: capsula articularis.

**Figure 18 animals-15-02894-f018:**
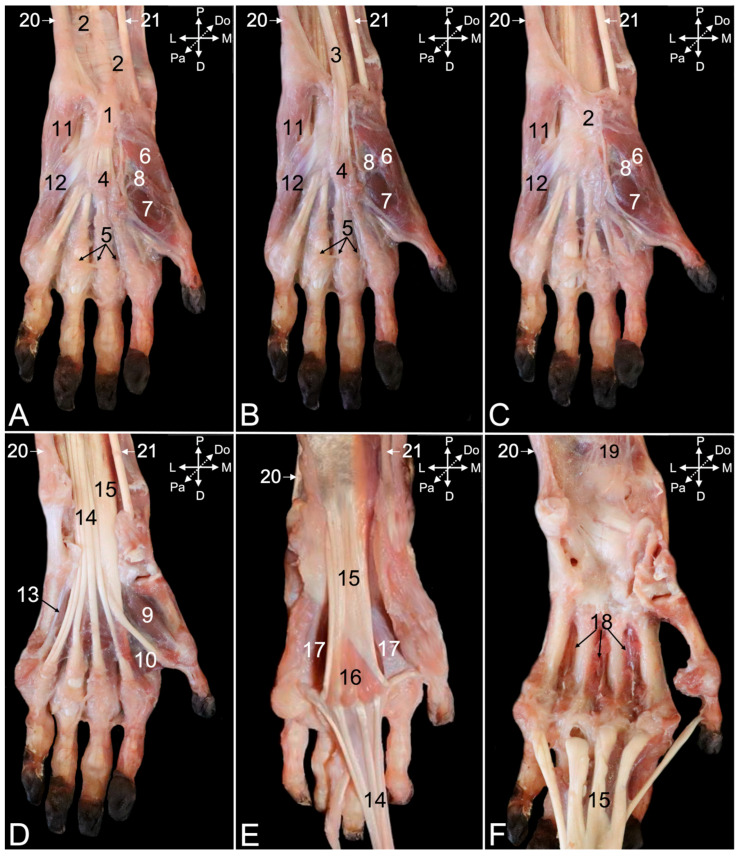
Palmar view of the musculature of the left hand of the hamadryas baboon: (**A**) superficial layer; (**B**) superficial layer after sagittally transecting the flexor retinaculum; (**C**) deeper layer after removing the palmaris longus muscle; (**D**) deeper layer after removing the deeper layer of the flexor retinaculum and the superficial intrinsic muscles of the first and fifth digits; (**E**) deeper layer after retracting the m. flexor digitorum superficialis in the distal direction; (**F**) deepest layer after retracting the m. flexor digitorum profundus in the distal direction, and removing the m. flexor carpi ulnaris and the m. flexor carpi radialis, and the palmar interosseous and contrahentes muscles of the hand. 1: fascia palmaris manus; 2: retinaculum flexorum; 3: m. palmaris longus; 4: aponeurosis palmaris; 5: lig. metacarpale transversum superficiale; 6: m. abductor digiti primi/pollicis brevis; 7: m. flexor digiti primi/pollicis brevis caput superficiale; 8: m. opponens digiti primi/pollicis; 9: m. flexor digiti primi/pollicis brevis caput profundum; 10: m. adductor digiti primi/pollicis; 11: m. abductor digiti quinti/minimi; 12: m. flexor digiti quinti/minimi; 13: m. opponens digiti quinti/minimi; 14: m. flexor digitorum superficialis; 15: m. flexor digitorum profundus; 16: mm. lumbricales manus; 17: mm. interossei manus palmares and mm. contrahentes digitorum manus; 18: mm. interossei manus dorsales; 19: m. pronator quadratus; 20: m. flexor carpi ulnaris; 21: m. flexor carpi radialis.

**Figure 19 animals-15-02894-f019:**
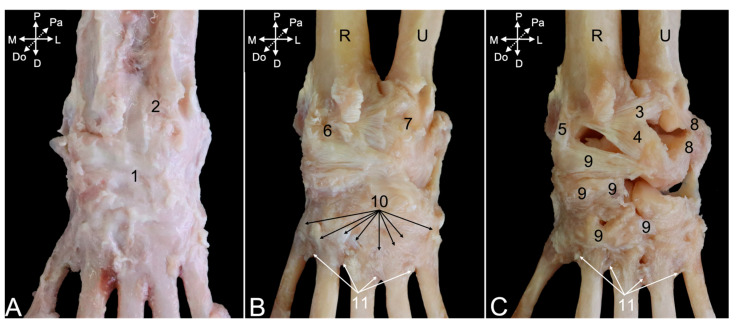
Dorsal view of the carpus and metacarpus of the left hand of the hamadryas baboon: (**A**) after removing the muscles and tendons showing the articular capsules; (**B**) after partial maceration of the soft tissues showing the ligament embedded in the joint capsules; (**C**) after additional maceration of the soft tissues showing the ligaments. 1: capsula articularis carpi; 2: capsula articularis radioulnaris distalis; 3: lig. radioulnare dorsale; 4: lig. radiocarpeum dorsale; 5: lig. collaterale mediale/radiale carpi; 6: capsula articularis radiocarpalis; 7: capsula articularis ulnocarpalis; 8: lig. collaterale laterale/ulnare carpi; 9: ligg. intercarpea dorsalia; 10: ligg. carpometacarpea dorsalia; 11: ligg. meta-carpea dorsalia; R: radius; U: ulna.

**Figure 20 animals-15-02894-f020:**
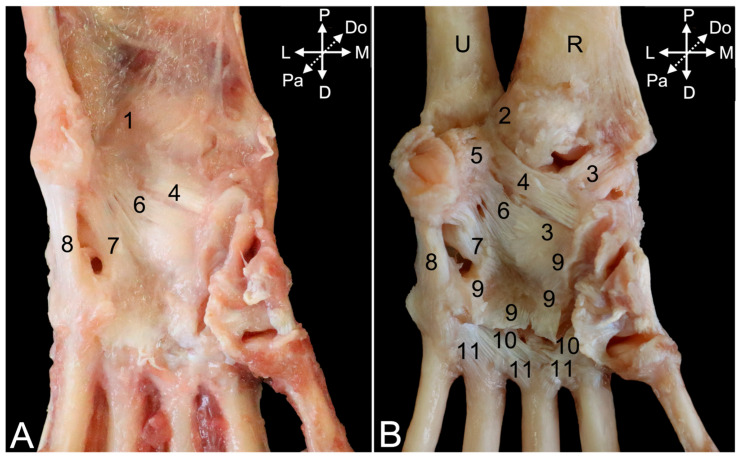
Palmar view of the carpus and metacarpus of the left hand of the hamadryas baboon: (**A**) after removing the muscles and tendons showing the articular capsules and some major ligaments; (**B**) after partial maceration of the soft tissues showing the ligaments. 1: capsula articularis radioulnaris distalis; 2: lig. radioulnare palmare; 3: lig. radiocarpeum palmare; 4: lig. ulnocarpeum palmare; 5: lig. ulnopisiforme; 6: lig. pisocentrale; 7: lig. pisohamatum; 8: lig. pisometacarpeum; 9: lig. carpi radiatum; 10: ligg. carpometacarpea palmaria; 11: ligg. metacarpea palmaria; U: ulna; R: radius.

**Figure 21 animals-15-02894-f021:**
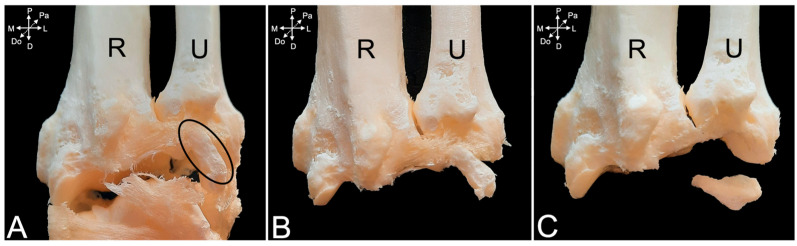
(**A**) Dorsal view of the left antebrachiocarpal joint of the male hamadryas baboon showing an additional carpal bone between the styloid process of the ulna and the triquetrum bone (encircled structure). (**B**) The structure is detached from the lateral collateral ligament and can be seen attached to the dorsal radioulnar ligament. (**C**) The structure is detached from the dorsal radioulnar ligament. R: radius; U: ulna.

**Figure 22 animals-15-02894-f022:**
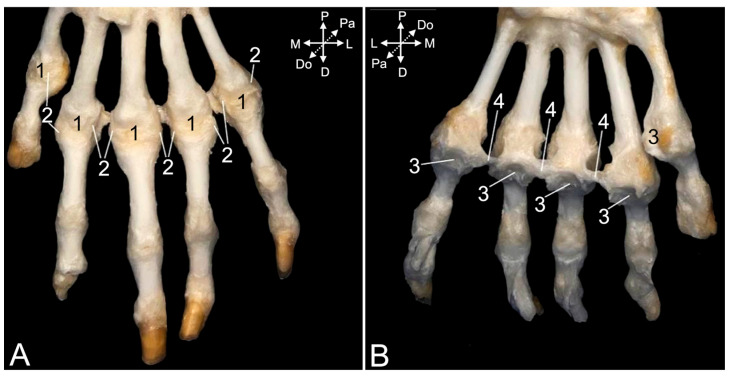
Dorsal view (**A**) and palmar view (**B**) of the ligaments of the left metacarpus of the hamadryas baboon. 1: capsula articularis metacarpophalangea; 2: ligg. metacarpophalangea collateralia; 3: ligg. metacarpophalangea palmaria; 4: ligg. metacarpalia transversa profunda.

**Figure 23 animals-15-02894-f023:**
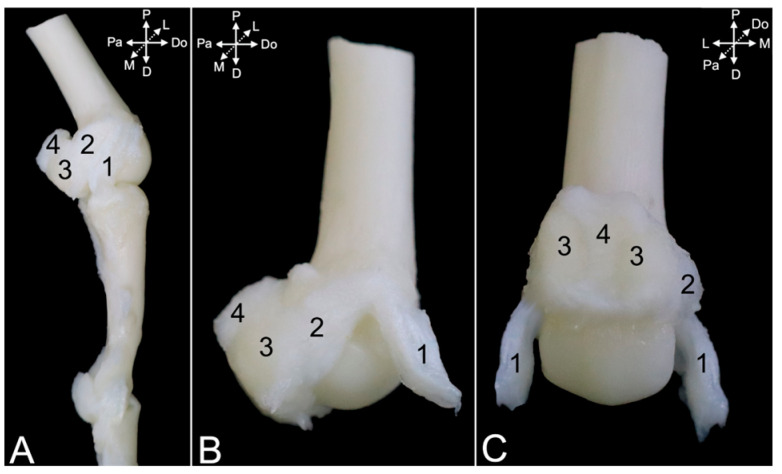
(**A**) Lower magnification of the axial view of the fourth metacarpophalangeal joint with its ligaments of the left hand of the hamadryas baboon; (**B**) higher magnification of the same view showing the same structures; (**C**) palmar view of the same structures. 1: lig. metacarpophalangeum collaterale; 2: palmar branch of the lig. metacarpophalangeum collaterale; 3: ossa sesamoidea palmaria proximalia; 4: lig. metacarpophalangeum palmare.

**Figure 24 animals-15-02894-f024:**
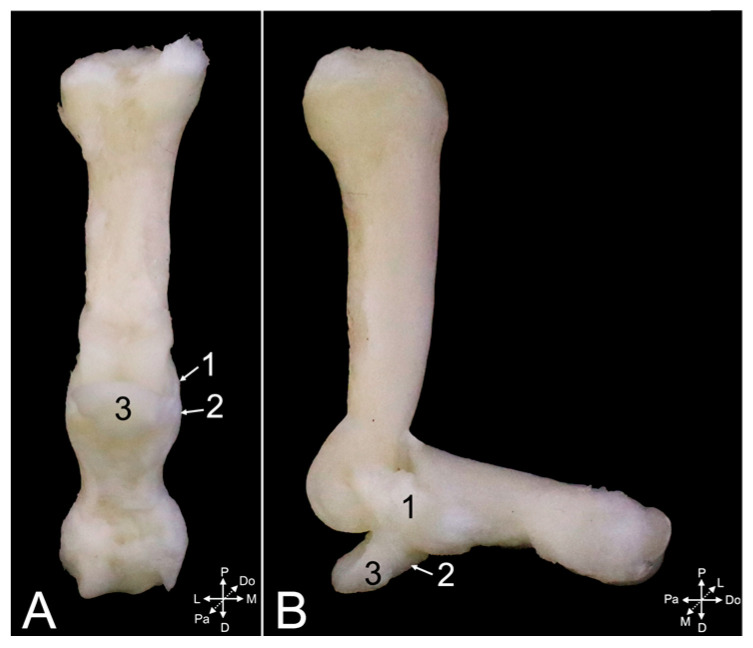
Proximal interphalangeal joint of the left fourth digit of the hamadryas baboon: (**A**) palmar view; (**B**) medial view. 1: lig. interphalangeum collaterale; 2: lig. interphalangeum palmare; 3: os sesamoideum palmare distale.

**Figure 25 animals-15-02894-f025:**
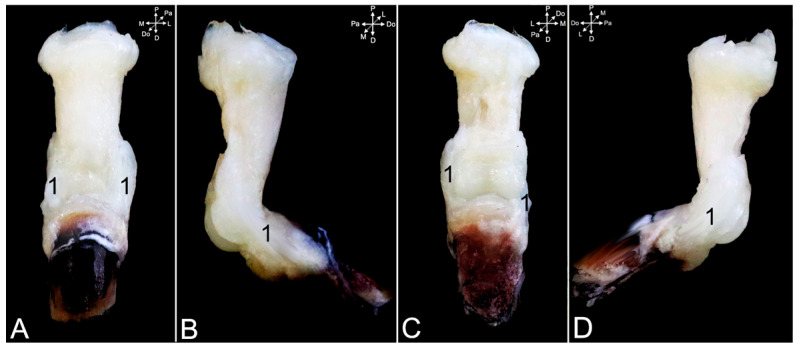
Distal interphalangeal joint of the left fourth digit of the hamadryas baboon: (**A**) dorsal view; (**B**) lateral view; (**C**) palmar view; (**D**) medial view. 1: lig. interphalangeum collaterale.

## Data Availability

Data are contained within the article.

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
