# Peer review of "Anatomy of the Joints in the Hamadryas Baboon (Papio hamadryas)—Part 1: Thoracic Limb"

_animals, 2025, doi:10.3390/ani15192894_

Round 1
Reviewer 1 Report
Comments and Suggestions for Authors
This manuscript aims to describe the upper limb anatomy of Papio using a dissection. Overall, it is an excellent, useful, and important work that shows the detailed anatomy of the cranial limbs. I know it is a lot of work to do this kind of study so I commend the authors. I have a few minor comments only.
In terms of references, there is no mention of Diogo and Wood's Comparative Anatomy and Phylogeny of Primate Muscles and Human Evolution (CRC Press, 2012), which includes some description of the upper limb musculature of a baboon and other monkeys. There is also no mention of Diogo's other works/atlases examining gorilla, bonobo, chimpanzee, and gibbon anatomy. Although these do not go into nearly a much detail as the current manuscript, they should be included in the reference list in the introduction and considered in the discussion.
In addition, some of the figures are difficult to understand due to a lack of orientation terms or guidance directly in the image or the caption. For example, Figure 1 is a lateral view of the left shoulder, but it's quite difficult to orient myself and locate the humerus, which is hidden under muscle in the first image. This is especially true if you are used to human anatomy books where the configuration is very different. I suggest adding at least one orientation term to each of the Figures, for example cranial, caudal, ventral, or dorsal, or adding a diagram that contextualizes the image, for example a diagram of the ribcage and limb in the orientation of the image.
Finally, some of the figures are a bit confusing because the numbers are larger than the structures they are labeling, or the structures are not clearly separated. For example, Figure 7 muscles 1, 2, and 3, as well as muscles 10 and 11, it is quite difficult to tell apart what is being designated by the numbers in part because the numbers cover most of the structure. It may be helpful to put the numbers outside of the muscle on the image and use an arrow, as was done for structure 14. It may also be helpful to include a diagram where the boundaries are clearly delineated. Something similar happens in Figure 10 in terms of difficulty identifying the structures, but rather because the text is white and there are a lot of white tendons and fascia visible which makes it difficult to identify the text on the muscles. I would suggest the authors go over each of the images to make sure the numbers are clearly legible and the structures clearly visible.
For figure 15, the circles and boxes are quite confusing. I think it would be more useful maybe to label the bones and then indicate which bones are included in each articular area? Or perhaps the bones could be shaded specific colors to conform to their articulations? For example, members of '8' could have the participating joint surfaces shaded white, and members of '5' could have participating joint surfaces shaded in red. Or at least have separate images for some of the articulations (for example, 1, 2, 3, 4, and 7 are all very difficult to tell apart) or to discern what is included in each group. The authors also mention earlier that a previous work focused on osteology, but it would be helpful for Figure 15 to include an image of the bones labeled, regardless of what choices are made in terms of labeling the articulations, so that the text can be followed more clearly.
I also did not notice any mention of variation between the specimens. Was there no variation? Or is an 'average' being presented? It would be useful to have statements in this regard, whether a general statement "We found no variation between individuals" or specific statements according to the findings.
Comments on the Quality of English LanguageThere are some minor areas in the introduction where the grammar could be improved.
Author Response
This manuscript aims to describe the upper limb anatomy of Papio using a dissection. Overall, it is an excellent, useful, and important work that shows the detailed anatomy of the cranial limbs. I know it is a lot of work to do this kind of study so I commend the authors. I have a few minor comments only.
In terms of references, there is no mention of Diogo and Wood's Comparative Anatomy and Phylogeny of Primate Muscles and Human Evolution (CRC Press, 2012), which includes some description of the upper limb musculature of a baboon and other monkeys. There is also no mention of Diogo's other works/atlases examining gorilla, bonobo, chimpanzee, and gibbon anatomy. Although these do not go into nearly as much detail as the current manuscript, they should be included in the reference list in the introduction and considered in the discussion.
We have added the suggested work together with The Visible Ape Project to the reference list and have cited these in the Materials and Methods section as valuable reference works during the dissections.
In addition, some of the figures are difficult to understand due to a lack of orientation terms or guidance directly in the image or the caption. For example, Figure 1 is a lateral view of the left shoulder, but it's quite difficult to orient myself and locate the humerus, which is hidden under muscle in the first image. This is especially true if you are used to human anatomy books where the configuration is very different. I suggest adding at least one orientation term to each of the Figures, for example cranial, caudal, ventral, or dorsal, or adding a diagram that contextualizes the image, for example a diagram of the ribcage and limb in the orientation of the image.
We truly understand this issue. We experience the same when consulting human anatomy atlases. We have put an orientation on each of the figures. When an isolated joint is difficult to orientate, we have additionally added a skeleton or skeletal structure.
Finally, some of the figures are a bit confusing because the numbers are larger than the structures they are labeling, or the structures are not clearly separated. For example, Figure 7 (now Figure 8) muscles 1, 2, and 3, as well as muscles 10 and 11, it is quite difficult to tell apart what is being designated by the numbers in part because the numbers cover most of the structure. It may be helpful to put the numbers outside of the muscle on the image and use an arrow, as was done for structure 14. It may also be helpful to include a diagram where the boundaries are clearly delineated. Something similar happens in Figure 10 (now Figure 11) in terms of difficulty identifying the structures, but rather because the text is white and there are a lot of white tendons and fascia visible which makes it difficult to identify the text on the muscles. I would suggest the authors go over each of the images to make sure the numbers are clearly legible and the structures clearly visible.
We have delineated the structures that were not easily differentiated. In some of the figures, we have put the labels in a more legible font.
For figure 15 (now Figure 16), the circles and boxes are quite confusing. I think it would be more useful maybe to label the bones and then indicate which bones are included in each articular area? Or perhaps the bones could be shaded specific colors to conform to their articulations? For example, members of '8' could have the participating joint surfaces shaded white, and members of '5' could have participating joint surfaces shaded in red. Or at least have separate images for some of the articulations (for example, 1, 2, 3, 4, and 7 are all very difficult to tell apart) or to discern what is included in each group. The authors also mention earlier that a previous work focused on osteology, but it would be helpful for Figure 15 to include an image of the bones labeled, regardless of what choices are made in terms of labeling the articulations, so that the text can be followed more clearly.
We did not really found a nice solution for this figure. This is because some smaller articulations are a part of a larger articulation, so there is overlap. However, have added an image of the hand skeleton with indication of the various carpal and metacarpal bones. This makes it much easier to follow the descriptions in the main text.
I also did not notice any mention of variation between the specimens. Was there no variation? Or is an 'average' being presented? It would be useful to have statements in this regard, whether a general statement "We found no variation between individuals" or specific statements according to the findings.
In the Discussion, we have added a few sentences on a variation we saw in the carpal joint. This was the only variation we have seen in the three dissected animals. We have stated this as such in the Discussion.
Reviewer 2 Report
Comments and Suggestions for Authors
Dear authors,
Your manuscript is a very good presentation of the detailed anatomy of the thoracic limb joints in Hamadryas baboon (Papio hamadryas). It is a meticulous, well-organized and well-presented work. I congratulate you for this research and presentation.
Each section is correctly presented both in terms of content and editing.
By studying this species whose anatomy is less described and by trying to compensate for the lack of veterinary terminology by adapting the terminology from NA, your work is a challenge to resume and establish the correct terminology of NAV for non-human primate species.
As a recommendation, it would be appropriate to specify whether the subjects of your study were euthanized or died of natural causes, and whether the cause of euthanasia or death did not influence the described morphology.
Author Response
Dear authors,
Your manuscript is a very good presentation of the detailed anatomy of the thoracic limb joints in Hamadryas baboon (Papio hamadryas). It is a meticulous, well-organized and well-presented work. I congratulate you for this research and presentation.
Each section is correctly presented both in terms of content and editing.
By studying this species whose anatomy is less described and by trying to compensate for the lack of veterinary terminology by adapting the terminology from NA, your work is a challenge to resume and establish the correct terminology of NAV for non-human primate species.
As a recommendation, it would be appropriated to specify whether the subjects of your study were euthanized or died of natural causes, and whether the cause of euthanasia or death did not influence the described morphology.
We have mentioned the reason for euthanasia and the method in the first paragraph of the Materials and Methods section.
Reviewer 3 Report
Comments and Suggestions for Authors
The study “Arthrology of the thoracic limb of the Hamadryas baboon (Papio hamadryas)” is of great scientific interest. The work is complete, thorough and very well documented. It offers the necessary knowledge for the care and interventions in Hamadryas baboons. The study is so well written that it leaves little room for improvement. I am impressed by the detail of the study and the clarity of the presentation.
Minor Comments/ Suggestions:
page 7, Figure 1: Letters/numbers on the photographs seem a little out of focus.
page 17, line 579: “know their insertions”. I think “know” is not the appropriate verb.
page 20, Figure 10: The photographs are too small. I think a little higher magnification would benefit understanding and make it easier for the reader.
page 23, line 774-775: “lateral collateral ligament can the oblique chord (chorda obliqua) (Figure 13B and insert, no. 6) that describes a diagonal, proximocaudal to distocranial course”. I think the sentence should be rewritten to be clearer.
page 26, Figure 15: I suggest that the outline of the antebrachiocarpal joint (2) be a little enlarged, change color (maybe orange) and the number 2 be repositioned for better viewing.
page 28, line 950: “Transecting it in the”. I think “it” should be replaced by “the flexor retinaculum” because the meaning of the sentence is not clear.
page 29, line 982-983: “At the level of the metacarpal bones, the tendon divides in four tendons that attach to the distal phalanges of all five digits”. Is this correct? The four tendons do not attach to the second to fifth digits?
Author Response
The study “Arthrology of the thoracic limb of the Hamadryas baboon (Papio hamadryas)” is of great scientific interest. The work is complete, thorough and very well documented. It offers the necessary knowledge for the care and interventions in Hamadryas baboons. The study is so well written that it leaves little room for improvement. I am impressed by the detail of the study and the clarity of the presentation.
Minor Comments/ Suggestions:
page 7, Figure 1: Letters/numbers on the photographs seem a little out of focus.
You are correct. We have relabeled this figure.
page 17, line 579: “know their insertions”. I think “know” is not the appropriate verb.
You are correct. This verb is not appropriate here. We have changed it to “have”.
page 20, Figure 10: The photographs are too small. I think a little higher magnification would benefit understanding and make it easier for the reader.
You are correct in stating that this figure is too small. We have made it larger using the entire width of the page.
page 23, line 774-775: “lateral collateral ligament can the oblique chord (chorda obliqua) (Figure 13B and insert, no. 6) that describes a diagonal, proximocaudal to distocranial course”. I think the sentence should be rewritten to be clearer.
We have rephrased and split the sentence. The sentence was indeed grammatically incorrect and too complex.
page 26, Figure 15: I suggest that the outline of the antebrachiocarpal joint (2) be a little enlarged, change color (maybe orange) and the number 2 be repositioned for better viewing.
We have worked on this figure to make it more understandable. We have added an image of the hand skeleton with indication of the various carpal and metacarpal bones. This makes it much easier to follow the descriptions in the main text. In addition, the original image has been upgraded by more and other colors and better positioning of the numbers.
page 28, line 950: “Transecting it in the”. I think “it” should be replaced by “the flexor retinaculum” because the meaning of the sentence is not clear.
Thank you for the suggestion. We have changed the phrase.
page 29, line 982-983: “At the level of the metacarpal bones, the tendon divides in four tendons that attach to the distal phalanges of all five digits”. Is this correct? The four tendons do not attach to the second to fifth digits?
It is not correct. Thank you for noticing. The common tendon splits in five tendons that attach to each of the five digits.
Reviewer 4 Report
Comments and Suggestions for Authors
Thank you very much for this interesting and necessary study on the thoracic limb joints of the hamadryas baboon. I share the authors’ view on the importance of species-specific anatomical studies. Below, I present some questions and suggestions regarding the manuscript:
I believe imaging exams are commonly used in the clinical routine for baboons. However, the authors did not include any radiographic images in the manuscript. Was this a deliberate choice, or was it not feasible to obtain such images? I believe it would be worthwhile to emphasize in the text the importance and necessity of this complementary method for visualizing bone structures and joints.
On that note, do the authors consider the lack of radiographic imaging a limiting factor in the study? What other limitations did the authors encounter during the research process? It might be valuable to include a discussion of these limitations in the manuscript.
The methodology section does not describe the dissection procedure for the skin and fascia. How was it performed—direction of the incision, reflection of the skin, etc.? Including these details would make the study even more specific and applicable as a dissection guide.
The authors refer to the connection between the thoracic limb and the thorax as a synsarcosis (line 137). How is this possible given that these animals possess a clavicle?
Are there photographs of the musculature involved in attaching the limb to the trunk (e.g., the trapezius)? It would be extremely helpful to include an image showing these muscles dissected in the whole cadaver.
The study provides a very detailed and rich description of the thoracic limb myology. Why is this not reflected in the title and objectives? The depth and detail regarding the musculature of this species’ thoracic limb truly stand out and deserve to be highlighted.
Finally, I congratulate the authors on the wealth of information and the detailed anatomical descriptions presented in this study. The manuscript is very well written and offers valuable comparative insights between human anatomy, non-human primates, and domestic animals.
Sincerely,
Author Response
Thank you very much for this interesting and necessary study on the thoracic limb joints of the hamadryas baboon. I share the authors’ view on the importance of species-specific anatomical studies. Below, I present some questions and suggestions regarding the manuscript:
I believe imaging exams are commonly used in the clinical routine for baboons. However, the authors did not include any radiographic images in the manuscript. Was this a deliberate choice, or was it not feasible to obtain such images? I believe it would be worthwhile to emphasize in the text the importance and necessity of this complementary method for visualizing bone structures and joints.
For sure it was possible to obtain radiographs because one of the authors works at the department of medical imaging. It was a deliberate choice not to include these. The manuscript is now 43 pages long and includes 25 often multi-panel figures. We think it would be better to have a separate manuscript focusing on medical imaging. We have provided a brief statement on the value of radiographs and suggest that this could be a topic for one of our future manuscripts.
On that note, do the authors consider the lack of radiographic imaging a limiting factor in the study? What other limitations did the authors encounter during the research process? It might be valuable to include a discussion of these limitations in the manuscript.
We have added a statement at the end of the Discussion that the low number of dissected specimens could be regarded as a limitation of our study. This fits in the anatomical variation we have encountered in the carpus of the male baboon. The lack of radiographs is not considered a limitation because we did not intend to investigate the joints using medical imaging. The focus of the present manuscript is on the gross anatomy and we think we have succeeded quite well. We have not mentioned this as such in the manuscript because we think this could be interpretated by the reader that it would be better to have radiographs, thus lowering the value of our manuscript. We do not think this is helpful.
The methodology section does not describe the dissection procedure for the skin and fascia. How was it performed—direction of the incision, reflection of the skin, etc.? Including these details would make the study even more specific and applicable as a dissection guide.
We have added the requested details on the dissection procedure in the Materials and Methods section.
The authors refer to the connection between the thoracic limb and the thorax as a synsarcosis (line 137). How is this possible given that these animals possess a clavicle?
You are absolutely correct. I think this lapsus has its origin in the fact that we, veterinarians, often work with animals that lack the clavicles. We have deleted the two passages dealing with the synsarcosis.
Are there photographs of the musculature involved in attaching the limb to the trunk (e.g., the trapezius)? It would be extremely helpful to include an image showing these muscles dissected in the whole cadaver.
We have added a new figure (Figure 1) that shows the muscles that attach the thoracic limb to the trunk.
The study provides a very detailed and rich description of the thoracic limb myology. Why is this not reflected in the title and objectives? The depth and detail regarding the musculature of this species’ thoracic limb truly stand out and deserve to be highlighted.
It is our intention to develop an anatomical atlas of the baboon and are currently investigating the joints. The muscular system is our next project. Therefore, we are deliberately not mentioning the musculature in the title or objectives.
Finally, I congratulate the authors on the wealth of information and the detailed anatomical descriptions presented in this study. The manuscript is very well written and offers valuable comparative insights between human anatomy, non-human primates, and domestic animals.